# Central pattern generator control of a vertebrate ultradian sleep rhythm

Lorenz A. Fenk[1,2,3 ✉], Juan Luis Riquelme[1,3] & Gilles Laurent[1 ✉]

The mechanisms underlying the mammalian ultradian sleep rhythm—the alternation of rapid-eye-movement (REM) and slow-wave (SW) states—are not well understood but probably depend, at least in part, on circuits in the brainstem[1–6]. Here, we use perturbation experiments to probe this ultradian rhythm in sleeping lizards (*Pogona vitticeps*)[7–9] and test the hypothesis that it originates in a central pattern generator[10,11]—circuits that are typically susceptible to phase-dependent reset and entrainment by external stimuli[12]. Using light pulses, we find that *Pogona*'s ultradian rhythm[8] can be reset in a phase-dependent manner, with a critical transition from phase delay to phase advance in the middle of SW. The ultradian rhythm frequency can be decreased or increased, within limits, by entrainment with light pulses. During entrainment, *Pogona* REM (REM$_P$) can be shortened but not lengthened, whereas SW can be dilated more flexibly. In awake animals, a few alternating light/dark epochs matching natural REM$_P$ and SW durations entrain a sleep-like brain rhythm, suggesting the transient activation of an ultradian rhythm generator. In sleeping animals, a light pulse delivered to a single eye causes an immediate ultradian rhythm reset, but only of the contralateral hemisphere; both sides resynchronize spontaneously, indicating that sleep is controlled by paired rhythm-generating circuits linked by functional excitation. Our results indicate that central pattern generators of a type usually known to control motor rhythms may also organize the ultradian sleep rhythm in a vertebrate.

Mammalian electroencephalographic (EEG) activity during sleep, described initially in humans and cats[13,14], consists of two main states: one characterized by slow-wave (SW) activity and the other being rapid-eye-movement (REM, also called active or paradoxical)[3]. Their alternation forms the so-called ultradian sleep rhythm. Studies in other mammals later established that, although a two-state description applies to most, the period, regularity, timing relative to the circadian cycle, relative fraction (or duty cycle) and coupling of the two states, internal structure and complexity (for example, EEG spectral contents, presence or not of sub-states) of sleep vary greatly across mammals[15], complicating the identification of common mechanistic principles. Studies in birds[16–19] and, more recently, in non-avian reptiles[7], provide evidence for SW- and REM-like activities also in non-mammalian amniotes. Although these similar activities may represent phenotypic convergence[20], they could instead reflect common amniote ancestry. If so, we reason that non-mammalian species, especially those evolutionarily closer to the stem ancestor, could help us identify ancestral, possibly shared, features of sleep control across all amniotes.

Although the mechanisms underlying mammalian SW–REM alternation are unknown[6], two classes of hypothesis exist, which rest on the identification of brainstem neurons with antiphasic activity during sleep, and on models inspired by these results. In the first class[1], monoaminergic neurons in the locus coeruleus (LC) and dorsal raphe inhibit cholinergic neurons in the pontine tegmentum with excitatory back projections. In corresponding models, these reciprocal interactions produce an alternating and sustained rhythm during sleep due to self-inhibition of the monoaminergic neurons releasing the cholinergic neurons to trigger a REM episode. Although these models have limit-cycle solutions qualitatively consistent with experiments[2], they rely on neural connections that have not been confirmed experimentally[4,6,21–23]. This led to a second class of models that rely on two key features, which together produce hysteretic loops[24]: the existence of mutually inhibitory circuits, suggested by anatomical data in the brainstem and elsewhere[4,24–26] to stabilize each sleep state; and the existence of a separate, slow-evolving and so-far-unknown mechanism, termed 'REM pressure', that accrues during waking state, SW activity or both, and triggers the transition between states[6,27].

Although different in their dynamic structures, these competing models agree on inhibitory feedback being a key circuit element to generate sleep-state alternation. Circuits that can generate alternating outputs without alternating input, often called central pattern generators (CPGs) are well known in motor systems[10,28,29]. For reasons laid out below, we wondered whether a CPG perspective might be useful to study the ultradian rhythm. We reasoned that (1) sleep's ultradian rhythm is probably the product of neural circuits rather than transcriptional regulatory loops as for the circadian rhythm; (2) REM production in mammals depends on the integrity of the brainstem/pons[1–4,6,25]; (3) in all vertebrates, the brainstem contains CPGs that control rhythms

[1]Max Planck Institute for Brain Research, Frankfurt, Germany. [2]Present address: Max Planck Institute for Biological Intelligence, Martinsried, Germany. [3]These authors contributed equally: Lorenz A. Fenk, Juan Luis Riquelme. ✉e-mail: lorenz.fenk@bi.mpg.de; gilles.laurent@brain.mpg.de

such as respiration, swallowing, whisking, singing, sighing, emesis and locomotion[11,30–34]; and (4) in the Australian dragon *Pogona vitticeps*, the biphasic sleep rhythm is, by virtue of its short period (2 min) and high regularity[7–9], consistent with the output of a CPG. We thus hypothesized that sleep's ultradian rhythm may be, at least in *Pogona*, the by-product of a CPG circuit, possibly evolutionarily related to brainstem circuits for motor control. Relying on extensive experimental, theoretical and computational studies of the shared properties of CPGs[10,12,28,29,35–37], we used phasic perturbations to probe the properties of *Pogona*'s ultradian sleep rhythm. Evidence for phase-dependent reset and entrainment of a neuronally generated ultradian rhythm, for example, would be consistent with the CPG hypothesis. Support for this hypothesis would, in turn, have helpful implications for our understanding of the evolution, development and mechanisms of sleep, especially because the genetic and developmental programs of pontine motor circuits are increasingly well understood in mammals[38,39].

We established previously[8,9] that the claustrum is an ideal recording site to identify the brain states of *Pogona* (Fig. 1a). Sleep in *Pogona* occurs at night and consists of equal-length epochs of SW and REM-like activity (REM$_P$) alternating every minute at room temperature for 8–10 h (ref. 7). Claustral SW is characterized by sharp-wave ripples (SWRs), generated by each claustrum independently[8,9]. REM$_P$, by contrast, contains stereotypical rapid and 'sharp negative' waveforms (SNs)[9] that occur irregularly about 20 times per second on average, thus generating local field potential (LFP) power in the 'beta' frequency range; SNs are generated in or upstream of the midbrain's isthmus and coordinated precisely across the two hemispheres via a winner-takes-all type of competition[9]. By measuring the LFP power in the beta range (12–30 Hz) in each claustrum (Fig. 1b, top), one can immediately identify the two phases of sleep (high beta during REM$_P$, low beta during SW) and the dominant side at any time (bottom, Fig. 1b).

## SW and REM$_P$ sleep alternate regularly

Shortly before the lights go out in the evening, the animals (being trained to a 12 h light/12 h dark rhythm) typically settle in one place, display decreasing postural tone and spontaneously start closing their eyes[7]. Figure 1c shows a sliding autocorrelation of the claustrum's LFP power in the beta band over 48 h. Black bars indicate the periods during which ambient lighting was turned off (nights, 19:00–07:00). The autocorrelation reveals the nocturnal ultradian rhythm (period, 122.5 s), characteristic of sleep in *Pogona*[7–9]. In the hour preceding dark, claustral activity displays fast, large-amplitude and irregular variations of beta-band activity (Fig. 1d and Extended Data Fig. 1a). This is characteristic of entry into sleep in *Pogona*. In the hour or two following dark, the irregular beta-band fluctuations become increasingly regular and settle into regularly alternating epochs of SW (roughly 0 beta power) and REM$_P$ (high beta power), with a period of 120–150 s (period mean ± s.d., 133 s ± 11 s; cycle count mean ± s.d., 218 ± 17; 20 animals) at room temperature (Fig. 1e and Extended Data Fig. 2). This periodic activity runs unaltered for 8–9 h, before returning progressively, at the end of the night, to an increasingly irregular and rapidly fluctuating state, similar to that preceding regular sleep (Extended Data Fig. 1a,b). When light returns (07:00), the animal typically opens its eyes, rapidly assumes an awake posture, and claustral activity becomes dependent on the animal's behaviour and activity (Fig. 1f). These features are typical of *Pogona* sleep (Extended Data Fig. 1a,b) and occur even when the lights are kept on throughout the night (Extended Data Fig. 1c) in our 12 h light/12 h dark-trained animals.

Noting the rapid periodicity of SW–REM$_P$ alternation during sleep, we suggest that this rhythm is due to the action of an oscillator circuit, gated by the circadian clock. If this is correct, this oscillator circuit might, once activated, express several hallmarks of a CPG: (1) a sensitivity to, and phase-dependent reset by, brief external stimuli and (2) entrainment by rhythmic drive at appropriate frequencies.

## Light pulses reset the ultradian rhythm

Sensory inputs such as light, sound or touch delivered to sleeping animals typically awaken them if sufficiently intense. We tested whether gentler stimuli could simply reset an on-going sleep rhythm without waking up the animals. Once a lizard reached its regular sleep in darkness, we imposed single 1-s-long ambient light pulses (mean 15.5 lx, Methods) delivered once every 30 min, at random phases of the sleep cycle (Fig. 1g, combined L and R beta power; Methods). These pulses caused no behavioural, myographic or encephalographic features suggestive of awakening nor did they cause eye opening (Extended Data Fig. 3 and Supplementary Video 1). They did, however, cause a reset of the on-going rhythm, as judged by the alignment of the next REM$_P$ onsets (at $t$ = 97 s, median; interquartile range (IQR), [91, 106] s; $n$ = 34 trials), independently of the phase of the light pulse in the on-going cycle. This reset is seen clearly in the averaged beta power over 34 trials and 3 nights: the mean power is flat before the light pulse, reflecting uncorrelated phases; it is oscillatory after (Fig. 1g, bottom trace), reflecting the alignment of beta activity caused by the light pulse. We varied the length of the light stimulus and observed a reset for durations as short as 10 ms (Fig. 1h, top), and as long as 90 s (Fig. 1h, bottom) (six recordings from five animals).

In animals sleeping in the dark, light pulses lasting many seconds often generated an increase in beta power, due to light-evoked SN trains (Fig. 2a), identical to those occurring during REM$_P$ (Fig. 2b). This is best seen when the light pulse fell during SW, when SNs were absent (Figs. 1h and 2a). This light-evoked REM-like activity ceased when the light pulse ended (30 s pulses, Fig. 1h), enabling SW to resume and a new REM$_P$ to follow some 80 s later (median, 81 s; IQR, [77, 93] s; $n$ = 77). With light pulses lasting longer than a normal REM$_P$ episode (for example, 90 s; Fig. 1h), beta activity very rarely reached the end of the light pulse (Figs. 1h and 2c; 3 of 66 trials, or 4.5%), suggesting that REM$_P$ duration is regulated internally and does not normally exceed a natural limit. Long light-pulse experiments also revealed that the reset was aligned with the light pulse (for example, 90 s pulses in Figs. 1h and 2c) rather than with the end of the light-evoked REM$_P$, implying that a full SW episode starts upon light-pulse offset, followed by REM$_P$ some 80 s later (median latency after 90 s light pulse, 77 s; IQR, [66, 83] s; $n$ = 66). In other words, REM$_P$ can be shortened by light stimuli but typically not extended beyond a natural limit, whereas SW can be both shortened and lengthened.

We next tested whether the effect of light pulses on the ultradian rhythm depended on their timing or phase (Fig. 2d,e). If a 1 s light pulse occurred in the first half of SW (Fig. 2d,e, blue), it lengthened SW, causing a phase delay. If it occurred in the second half of SW (Fig. 2d,e, green), it triggered an early (and usually short) REM$_P$, causing a phase advance. If, by contrast, the light pulse fell during REM$_P$, its effects were minor (Fig. 2d, orange, red; Fig. 2e, phase response curve). This pinpoints the middle of SW as a critical phase of the sleep cycle, during which the effect of an external stimulus on the on-going rhythm switches abruptly from a phase delay to a phase advance. A 90-s-long light pulse also had a phase-dependent effect (Fig. 2f, black), although it differed slightly from that for 1 s pulses (and the reset was pushed to 70–80 s after the end of the pulse). In summary, reset of the ultradian sleep rhythm by a pulse of light is phase dependent.

## Sleep-rhythm entrainment by light pulses

The intrinsic frequency of a CPG is usually adapted to the resonant properties of the physical device (for example, a limb) that it controls[40]. CPG and limb thus often act as coupled oscillators, each influencing the other through command and feedback, respectively, to shape a stable combined output. Consequently, rhythmic external inputs to a CPG are generally best at entraining it if their frequency is near the natural frequency of the CPG. This can cause a shift in the CPG's on-going

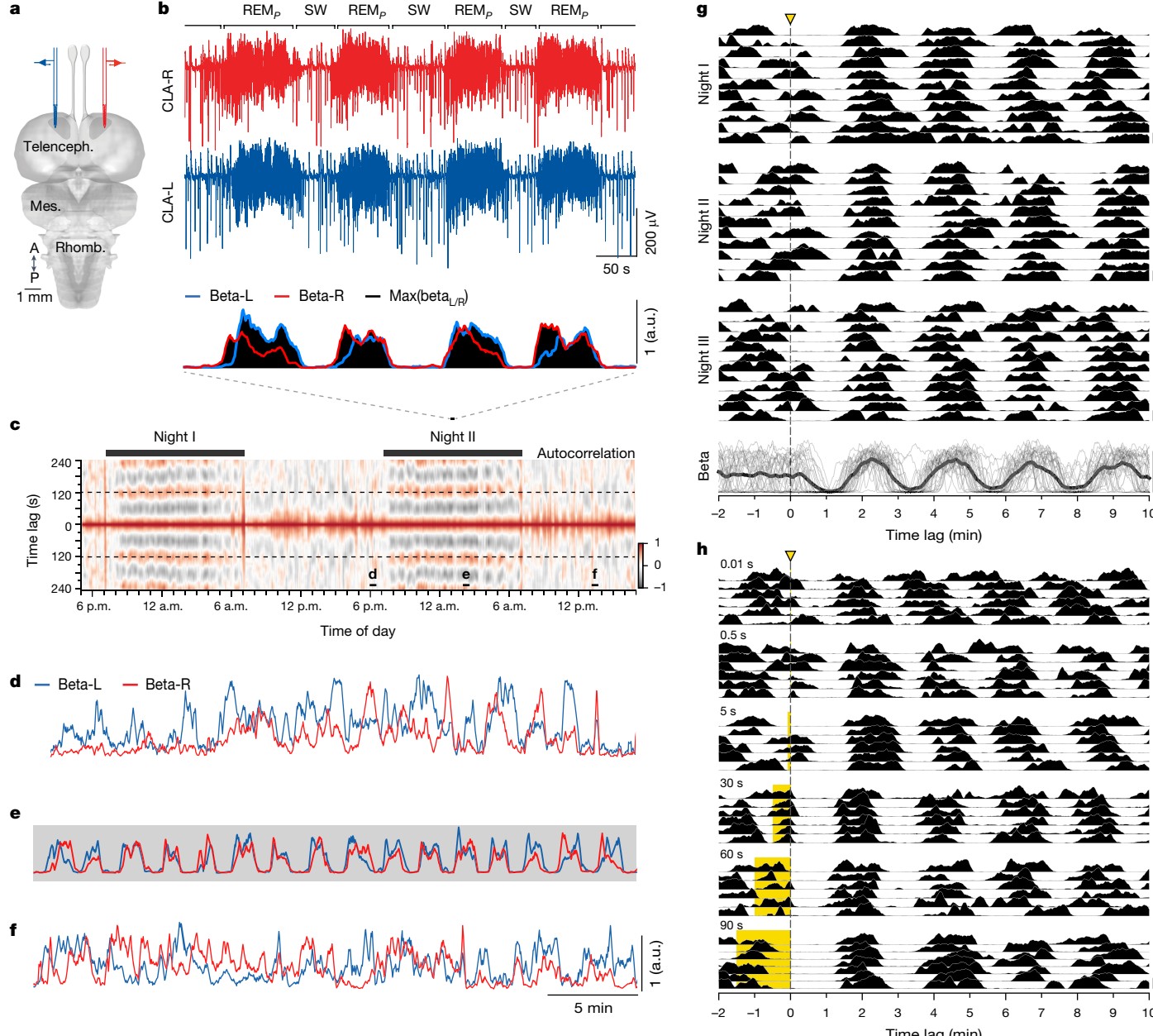

**Fig. 1 | *Pogona*'s ultradian sleep rhythm is reset by light pulses. a**, Schematic of the *Pogona* brain with electrodes in the claustra. **b**, Top two traces, LFP from both claustra (left, CLA-L and right, CLA-R) in the middle of the night, with four REM$_P$ bouts, interspersed with SW (containing SWRs). Bottom, scrolling power in the beta band in each claustrum, and their maximum (max; black). Note dominance switches during REM$_P$ (ref. 9). **c**, Scrolling autocorrelation of the max power of the LFP in the beta band, revealing ultradian rhythm (approximately 2 min period, stippled lines), for about 8 h each night. **d–f**, Half-hour segments from **c**, showing the temporal variations of beta power in each claustrum. Note regularity in **e** (shading indicates lights off). Full 24 h recordings from three animals are shown in Extended Data Fig. 1. **g**, Single 1 s light pulses (stippled line) delivered to the closed eyes of one sleeping animal over three nights reset its ultradian rhythm, estimated from variations of beta band max power, as in **b**. Trials every 30 min over three nights. Because light delivery is not locked to the ultradian rhythm, average beta power (bottom trace, black) oscillates only after pulse delivery, reflecting the reset. Waning of the averaged oscillation indicates noise in the ultradian rhythm frequency: after 30 min, a new light pulse is not locked to the rhythm, ensuring that sampled phases are distributed randomly. **h**, Effects of light-pulse duration (10 ms to 90 s). A 10 ms pulse causes a reset, but with low reliability. Long pulses (30 s and above) often trigger a REM$_P$ episode that ends with the light pulse. Pulses longer than a normal REM$_P$ cycle fail to lengthen REM$_P$ (90 s) beyond its natural duration, yet the start of a new REM$_P$ cycle is aligned to the offset of the light pulse. Pulses aligned at offset; six nights, five animals. A, anterior; P, posterior; a.u., arbitrary units; Telenceph., telencephalon; Mes., mesencephalon; Rhomb., rhombencephalon.

frequency if it is active, or the forced activation of a silent CPG into a rhythmic state, which may persist even after the periodic forcing stimulus has been withdrawn. We found evidence for both.

To test for entrainment, we first presented trains of 1-s-long light pulses to sleeping animals at rates deviating from their intrinsic ultradian frequency (140 s ± 9 s across 65 non-stimulated periods from

4 animals). Using inter-pulse intervals (IPIs) shorter or longer than the period of its natural rhythm (135 s ± 6 s for the animal in Fig. 3a), we could accelerate or slow down an animal's sleep rhythm with IPIs between 120 and 200 s (Fig. 3a–c), thus forcibly reducing or increasing the sleep-cycle duration. Intervals outside of this range (for example, 80, 100 and 240 s; Fig. 3a) failed to entrain reliably (for example, failure

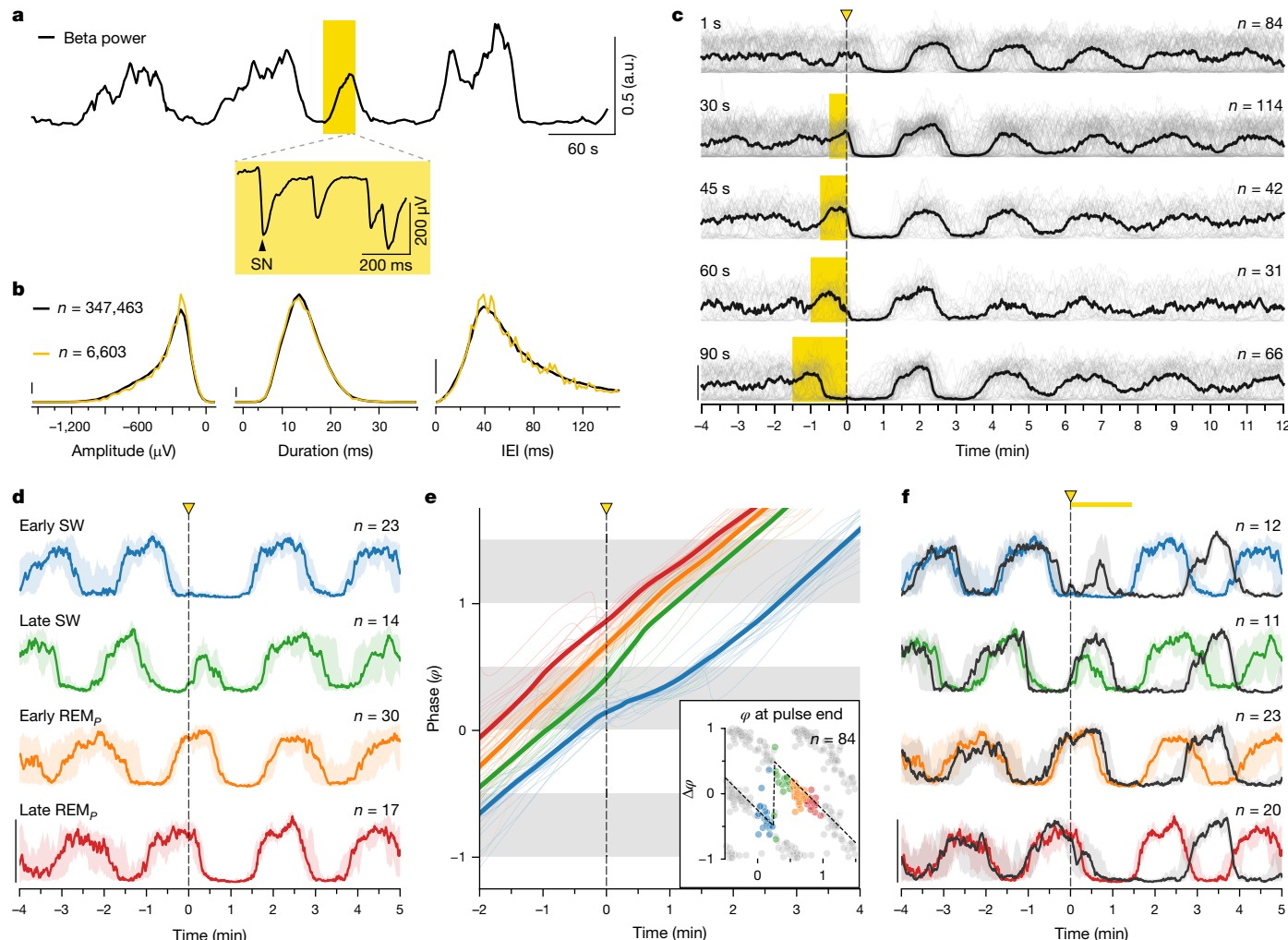

**Fig. 2 | Reset of the ultradian rhythm is phase dependent. a**, A light pulse occurring near the middle of an on-going SW triggers an early episode of REM$_P$, characterized by SNs (inset) and a concomitant increase in beta power. **b**, SNs evoked by a light pulse (yellow) are identical in amplitude, duration and inter-event interval (IEI) to those during normal REM$_P$ (black). **c**, With sustained light pulses, reset is aligned to pulse offset. With pulses 60-s-long and over, the light-triggered REM$_P$ cycle ends when it reaches its normal duration, that is, before the end of the light pulse. Ordinate, power in beta band; grey, individual trials; black, medians; 14 animals. **d**, Beta power (medians) for trials grouped by phase of light pulse onset (1-s-long pulse). Shading indicates IQR; four animals; *n* indicates trial number. A pulse in early SW causes a delay (blue). A pulse in later SW causes an early (and rapidly aborted) REM$_P$ (green). **e**, Phase response analysis of single (thin) and average (thick) trials in **d**. Main, unrolled phase (in *y*) of beta power relative to light onset (*t* = 0). Note divergent inflexions of blue and green curves at *t* = 0. Inset, phase response curve for the four groups of trials. Grey dots are mirrored data. Note sharp switch from delay (negative Δ*φ*, blue) to advance (positive, green) when pulse moves from early to late SW (stippled line). **f**, Same as **d**, but including trials with 90 s light pulses (dark grey); three animals. Note similar effects, but with a reset delayed by 90 s. Coloured traces as in **d**, for comparison.

after the third pulse at IPI = 240 s; Fig. 3a). For IPIs causing reliable entrainment (120–200 s), the distribution of pulse phases was unimodal and narrow (0.95 at 160 s, corresponding to the REM$_P$–SW transition; Fig. 3b), indicating phase-locking of stimulus and cycle (160 s; Fig. 3c). For IPIs with poor entrainment, the distribution displayed high variance (240 s; Fig. 3b) with pulses producing a mixture of phase advances and delays (240 s; Fig. 3c). Note that the IPI causing best locking (160 s) was slightly longer than this animal's natural sleep-cycle period (130–140 s) and that, with long IPIs generating unreliable locking (for example, 220 or 240 s), the timing was such that the light pulse often fell near the middle of SW, when the consequences on reset (phase delay or advance) are most sensitive (Fig. 2e): it often caused a phase delay, seen as a dip in the average phase versus time curve (Fig. 3c, cyan, bottom two panels), similar to what is seen with single pulses delivered during early SW (Fig. 2d,e). Slowing down the ultradian rhythm could be achieved over a wider range of imposed periods than speeding it up (period increase: +40% versus period decrease: −14%; Fig. 3d). These effects of entrainment on period were caused mainly by an elongation

of SW (Fig. 3f and Extended Data Table 1); REM$_P$ duration remained bounded to its maximum natural duration over the range of entraining IPIs (Fig. 3e and Extended Data Table 1). When the IPI was very short (80 s), the extension of SW (Fig. 3f) was accompanied by a concomitant reduction or even suppression of REM$_P$. This suppression was typically followed by a rebound of REM$_P$ and a reduction of SW after the stimuli ceased (Extended Data Fig. 4). The greater elasticity of SW compared with that of REM$_P$ is consistent with the results in Fig. 2 (note the asymmetry around *φ* = 0.25 of the phase response curve; Fig. 2e, inset).

## Entrainment in awake animals

We next tested whether an external periodic stimulus could trigger an ultradian-like rhythm in awake animals; this would be a strong indication of the activation, and thus existence, of a pattern generator. We exploited the fact that the diagnostic components of claustral activity during sleep (SWRs during SW, SNs during REM$_P$) are also expressed in awake animals submitted to a sudden transition from

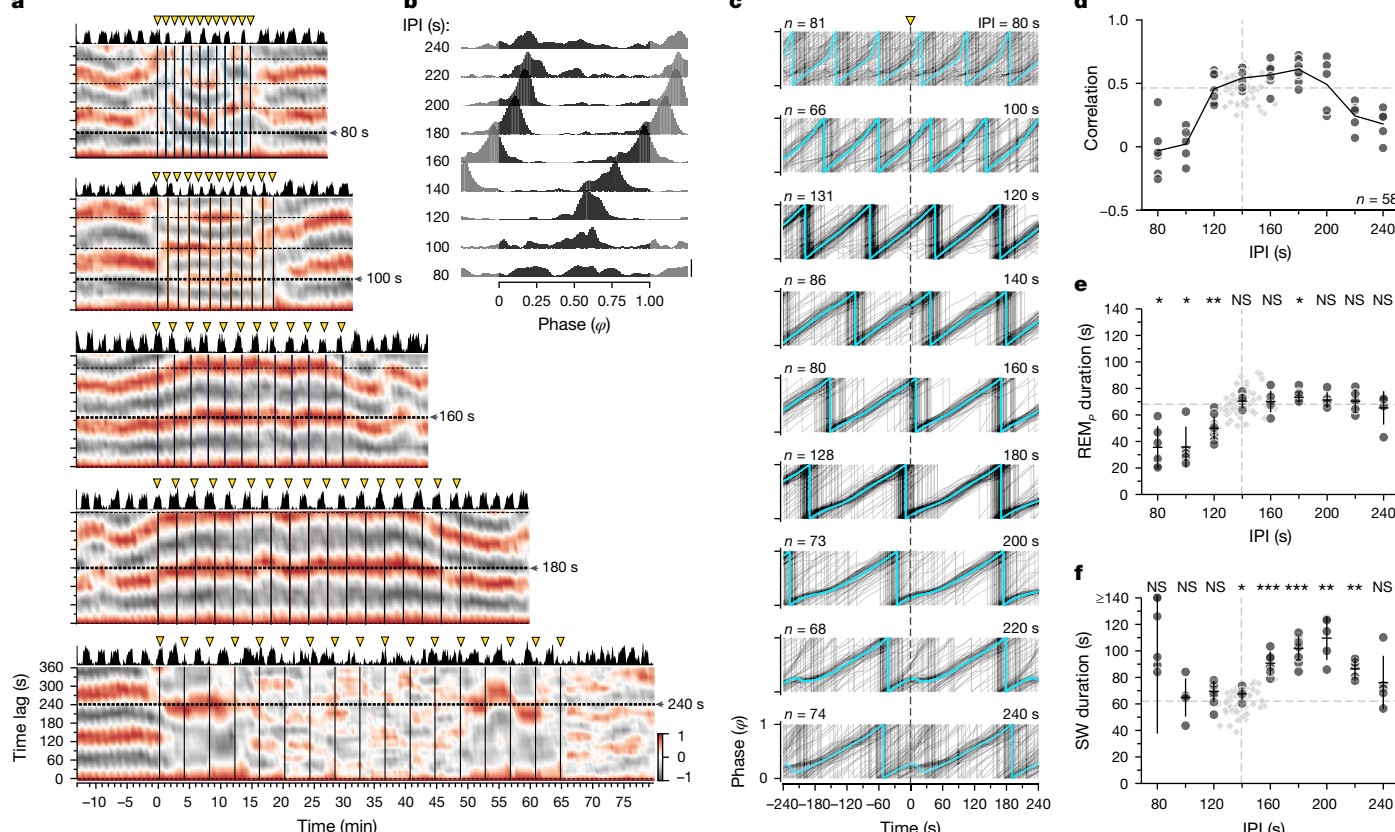

**Fig. 3 | Sleep's ultradian rhythm can be entrained by trains of light pulses.**
**a**, Sliding autocorrelation of beta power (REM_P). Pearson's correlation, for five inter-pulse intervals (IPIs, 80–240 s). Triangles, 1 s light pulses. Beta power at top. Natural cycle period for this animal, 135 s. Entrainment occurred (horizontal bands) with IPIs of 100, 160 and 180 s, but failed with shorter and longer IPIs. **b**, Distributions of pulse phase during entrainment. Scale bar, density of 1. 81–131 pulses per IPI. Narrow distributions indicate phase-locking and good entrainment (120–200 s IPIs). **c**, Locking precision varies with IPI, with a peak around 160–180 s, hence slightly longer than the natural period. $t = 0$, time of light pulse. Cyan, circular means. **d**, Circles, autocorrelation of beta power (*y*) at lag corresponding to the imposed IPI (*x*). Line joins means. Correlation drops for short and long IPIs, as in **a**. Diamonds, statistics of spontaneous sleep rhythm in the 25 min preceding each pulse train.

In *x*, autocorrelation period; in *y*, autocorrelation values. Stippled lines, mean *x* and *y* for baseline measurements. **e**, Relationship between IPI and median REM_P duration. Black lines indicate mean and s.d. Diamonds, median REM_P duration (*y*) in the 25 min preceding one pulse train (*x*). REM_P duration does not increase beyond its spontaneous value for IPIs where entrainment occurs. NS, not significantly different from distribution of baseline sleep values (diamonds). *$P < 0.1$; **$P < 0.05$. Two-sided paired *t*-tests (statistics in Extended Data Table 1). **f**, Same as **e** for SW duration. SW duration increases as IPI increases, up to IPIs where entrainment fails (≥220 s). Values above 140 s (80 s IPI, see EDF4a) are clipped. *, **, as in **e**; ***$P < 0.001$. Data shown in **b**–**f** are from 4 animals (12 nights). Two-sided paired *t*-tests (statistics in Extended Data Table 1).

light to dark (causing SWRs) or from dark to light (causing SNs; Fig. 4a). Under these conditions, claustral SWRs and SNs are indistinguishable from those observed during SW or REM_P, respectively (Fig. 4b). These experiments were run around the middle of the day, when the animals were awake, upright and attentive (30 experiments from 10 animals). At no time did the animals lose nuchal, axial or limb muscle tone (Supplementary Video 2), characteristic of sleep[7]. In particular, every time the light was turned on, the animals opened their eyes if they had been closed—a reaction to light pulses never observed in sleeping animals.

Upon turning the lights off, claustral beta power dropped (Fig. 4a, i–ii transition), SWRs appeared and the animal closed its eyes with a latency of seconds to tens of seconds (Fig. 4a(ii, iii),b). SWR production (mimicking SW) could be maintained for up to 90 s in artificial darkness (Fig. 4a). SWRs could start before (Fig. 4a(ii)) or after eyelid closure; hence, eyelid closure was not the cause of the SWRs. If darkness was maintained longer, SWRs became less frequent and SNs—typical of open-eyed awake states and of REM_P—appeared, whether the eyes were open or closed (Fig. 4a(iv),b). When light returned, SNs replaced SWRs if the latter had not already ceased, and beta power rose again. Thus, a regime of alternating light and dark pulses with

appropriate duration and interval produced alternating epochs of SNs (high beta power) and SWRs (low beta power) (Fig. 4c), mimicking what occurs spontaneously during sleep (Fig. 1e). We tested whether this stimulation regime could activate a putative ultradian oscillator from a 'dormant' state. If so, entrainment at the appropriate frequency might cause it to remain active even after the alternating light/dark stimuli had stopped.

We used alternating OFF–ON light pulses repeated 9–13 times every 30, 45, 60, 80 or 90 s, generating an equal number of cycles of alternating SW-like (lights off) and REM_P-like (lights on) activity in the claustrum (Fig. 4c–g) (15 experiments in 8 animals, for 60 s pulses; Fig. 4c). At the end of this train, we turned the light off once more and kept it off (Fig. 4c–g). Although the animal was then in the dark, beta power rose again when the next light pulse should have occurred, but did not (60 s in Fig. 4c,f,g; with 60 s entraining pulses: median entrained beta onset, 61.5 s; IQR, [59.7, 68.9], $n = 15$; 90 s entraining pulses: median entrained beta onset, 82.0 s; IQR, [80.3, 95.5]; $n = 5$; single example in Fig. 4d). This REM_P-like activity ended on its own within the variance of a normal REM_P epoch, and could start again once or twice more at the appropriate frequency but with waning vigor (Fig. 4d,f) even though no light pulse was provided. This manipulation never led to a persistent

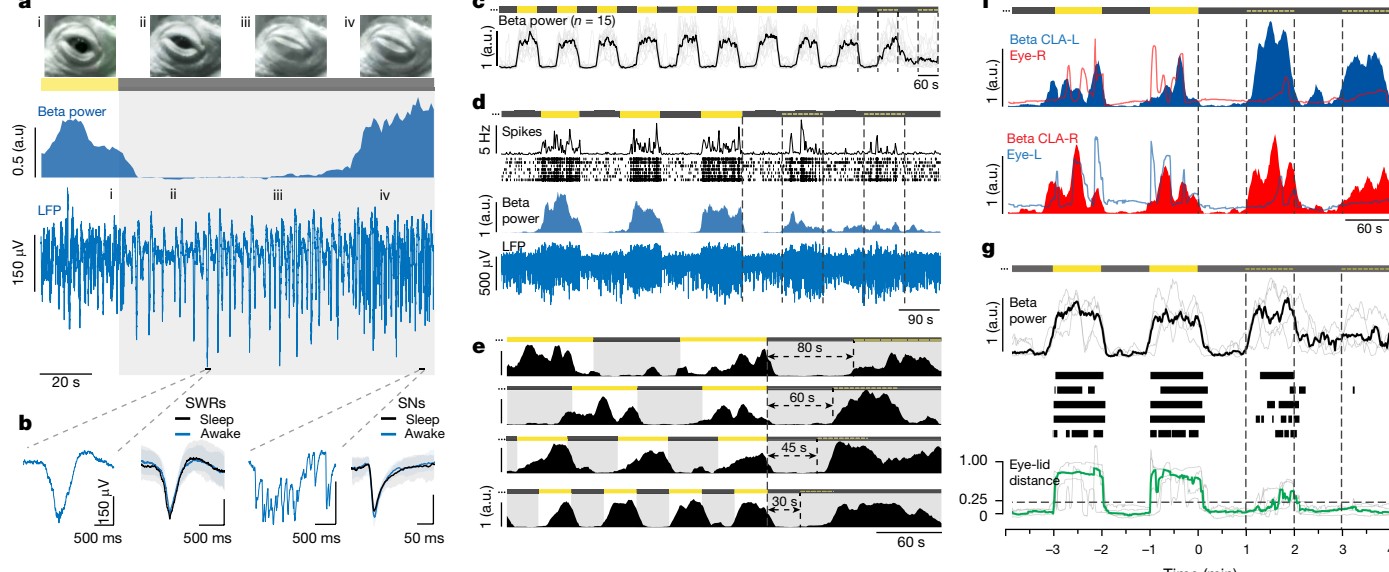

**Fig. 4 | A sleep-like rhythm can be entrained in awake animals. a**, Light off (grey) caused switch from SNs (i) to SWRs (ii, iii) and delayed eye closure (iii). After 60–80 s, SNs return, even though animal is awake, in the dark with eyes closed (iv). LFP 0.25–100 Hz band-pass. **b**, SWRs and SNs evoked in awake animals are identical to those in sleep. First and third panels: single traces from **a**. Second and fourth panels: blue, median and IQR of SWRs ($n = 100$) and SNs ($n = 100$) during awake entrainment; black, same during sleep (same animal). **c**, After 9–13 alternating dark/light, each 60 s (9 shown), light is left off. Beta power returns when light is expected (stippled yellow line). Black, median of 15 trials; 8 animals. **d**, Same as **c**, with 90 s period for one animal. From bottom, claustral LFP; beta power; CLA units and instantaneous rate; REM$_P$-like activity occurs when light should have returned. **e**, End of awake entrainment regime

with four periods (30–80 s, one animal); whereas entrainment occurs with all four periods, beta activity does not return spontaneously after less than 60 s. **f**, Entrained activity in darkness does not depend on eye opening. Beta power in L and R claustra, and contralateral eyelid distance (lines); entrained beta bursts begin before eyes open, briefly if at all. **g**, Awake entrainment (four animals, five days) showing, from bottom, distance between eyelids (green, median; grey, trials), eye openings (thresholded eyelid distance), beta power (black, median; grey, trials) and last two pulses of light entrainment. Note late eye openings relative to entrained beta (at $t$ = around 1.5 min). Numbers of experiments, 3 30 s pulses in 2 animals; 3 45 s pulses in 3 animals; 15 60 s pulses in 8 animals; 1 70 s pulse in 1 animal; 1 80 s pulse in 1 animal; 5 90 s pulses in 4 animals.

rhythm. Entrainment occurred with several light/dark frequencies, but the entrained response matched the entrainment protocol only if the latter fell within the natural range of the ultradian rhythm (Fig. 4e). For example, with entrainment periods of 160 and 120 s (Fig. 4e, first and second traces), REM$_P$-like activity reappeared at the expected +80 and +60 s (half-periods). But with shorter (thus unnatural) periods, REM$_P$-like activity reappeared after about 60 s (Fig. 4e, bottom two traces), confirming the lower bound observed with entrainment during sleep (Fig. 3f). The entrained REM$_P$-like epochs were not the result of eyelid opening at the time when light was expected: REM$_P$-like activity started in the dark well before eye opening, if eye opening happened at all (Fig. 4f,g). In conclusion, an ultradian-like rhythm can be activated for a brief time by appropriate light/dark entrainment in awake animals, consistent with the hypothesis that the ultradian rhythm is under the influence of an oscillator circuit, normally inactive in awake animals.

## A pair of coupled oscillators

In vertebrates, all known CPGs (for example, respiration, vocalization and locomotion) located in the hindbrain or spinal cord, are paired[11,30–33,38,41]. In mammals, control for REM–SW alternation is thought to reside, at least in part, in the pontine region[1,4,6,26,42]. Thus, if the ultradian rhythm is produced by a pontine CPG, we might expect it to be paired. In *Pogona*, we recently characterized a winner-takes-all competition between left and right hemispheres during REM$_P$ (ref. 9) that depends on bilateral isthmic nuclei, in a region just anterior to the pons. We therefore examined whether our hypothesized ultradian CPG is, like all known vertebrate CPGs, also paired. The optic nerves in *Pogona* undergo a complete decussation (Extended Data Fig. 5), so that a retinal output reaches each hemisphere (optic tectum and

thalamus) from the contralateral eye. In all the above sleep experiments (Figs. 1–3), reset and entrainment were generated by light acting on both retinae through closed eyelids: covering both eyes with opaque cups (Methods) suppressed the effects of light (Extended Data Fig. 6). Here we occluded one eye (Fig. 5a), recorded from both claustra and repeated the reset experiments (in sleeping animals, at night, in the dark), as shown in Fig. 1. A single light pulse had the same immediate effect on the ultradian rhythm as described above (phase-dependent reset), but only in the claustrum contralateral to the uncovered eye ('seeing' claustrum; Fig. 5a,b). In the 'blind' claustrum, the one linked to the covered eye, beta/REM$_P$ activity continued without phase shift (Fig. 5c,d). Yet, both sides eventually re-synced (Fig. 5e,f). Re-syncing could happen early, that is together with the reset of the 'seeing' claustrum (Fig. 5f, top), as seen in binocular experiments; the two sides then continued in synchronized sleep mode. In other cases, the re-synchronization occurred at a later sleep cycle, typically no later than with the fourth REM$_P$ cycle following the light pulse (Fig. 5f, bottom three traces). How quickly both sides re-synced depended (with some noise) on the phase of the sleep cycle at which the monocular light stimulus fell (Fig. 5g)—a pulse falling near the midpoint of SW ($\varphi = 0.25$) led, on average, to the longest delays to re-synchronization, consistent with the fact that this phase corresponds to the largest $\Delta\varphi$ spread (Fig. 2e, inset and Fig. 5b). The unilateral resetting combined with later re-synchronization, however, show that two ultradian oscillators exist (one per side) and that they are coupled by net excitation.

## Discussion

*Pogona*'s fast-paced and regular ultradian sleep rhythm could be reset by ambient light pulses delivered to closed eyelids in a phase-dependent

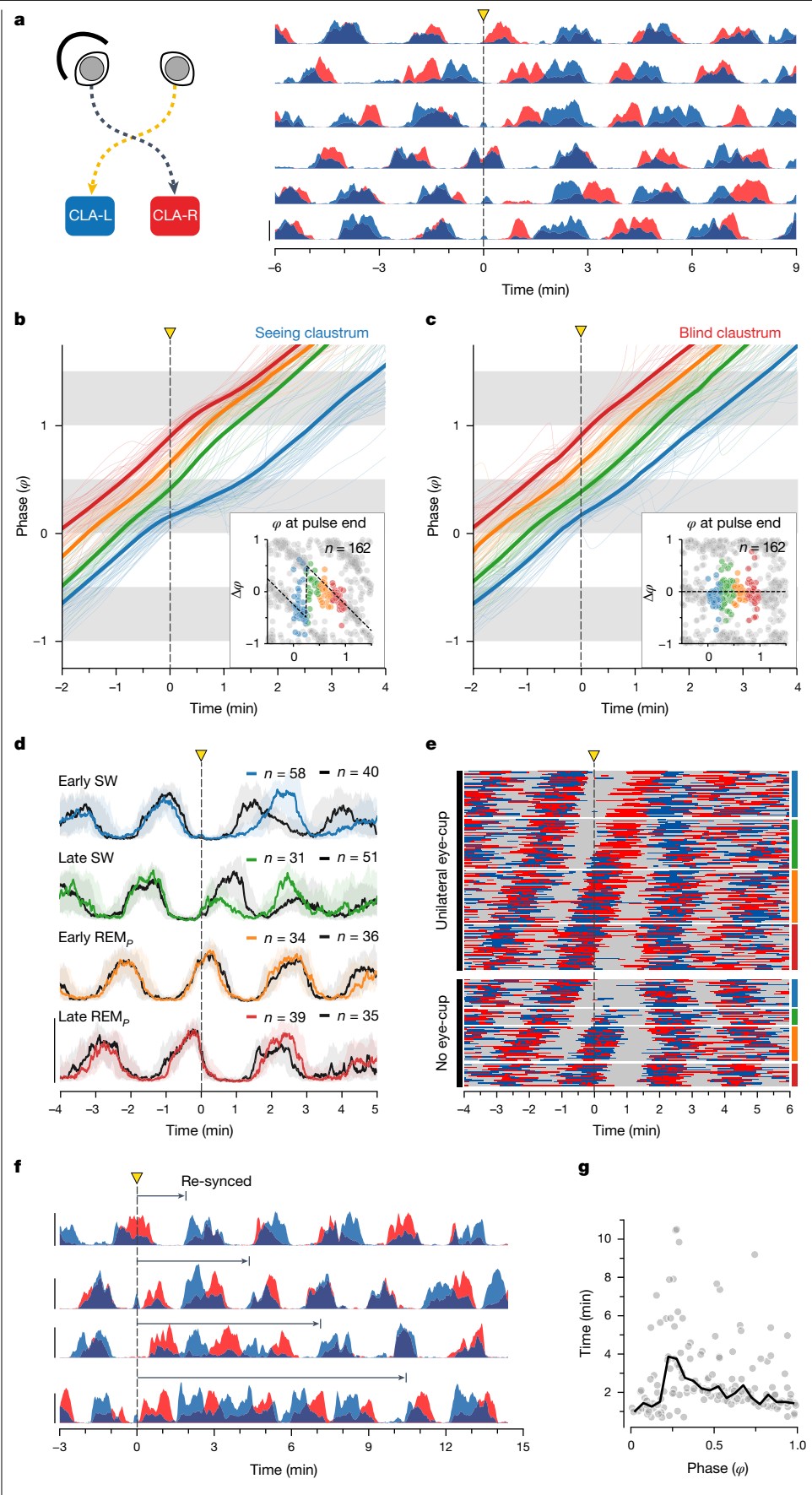

**Fig. 5 | Reset is unilateral if light affects a single eye. a**, Sleeping animal, both eyes closed, one eye (L in schematic) occluded with a cover. Individual trials with 1 s pulses delivered at random phases of the ultradian rhythm. Only claustrum contralateral to the seeing side (blue) experiences a reset. Both sides resynchronize after a few cycles (**e**–**g**). **b**, Unrolled phase of beta for 'seeing' claustrum. Single (thin) and mean (thick) trials grouped by phase of light pulse (blue, early SW; green, late SW; orange and red, $REM_P$). Responses as in binocular reset (Fig. 2e); four animals. Inset, phase response curve. **c**, Same as **b** for 'blind' side shows no phase reset. **d**, Overlay of the responses of the two sides ($n = 162$ trials) to a 1-s-long light pulse: black, beta power from 'blind' claustrum; colours, beta power from seeing claustrum grouped by phase of light pulse, as in Fig. 2d. Medians and IQR. **e**, Comparison of effect of light pulse with (top, $n = 162$) and without (bottom, $n = 84$) unilateral eye cup. Trials grouped by phase of light pulse (colour groups as in **d**). Grey indicates SW (low beta power). Red (right, blind claustrum) and blue (left, seeing claustrum) lines represent $REM_P$ in dominant claustrum (that with greatest beta power) at any time. Note that blind side (red) experiences no reset in unilateral eye-cup trials (top). Reset is visible as interruption of central $REM_P$ diagonal on trials with no eye-cup (bottom). **f**, Although only the seeing side (blue) experiences immediate reset, the two sides re-sync after one to four sleep cycles. **g**, Time of bilateral re-syncing (*y*) against phase of light pulse. Longest delays around $\varphi = 0.25$ (SW midpoint). Black, rolling median. From 162 1-s-long pulses; 13 nights, 4 animals.

manner, and entrained at frequencies higher or lower than its natural frequency. In awake animals, an ultradian-like rhythm could be revealed briefly by entrainment, using alternating light and dark periods of durations matching those of natural $REM_P$ and SW. This awake manipulation did not cause sleep: it only entrained a circuit whose output matched that of the ultradian rhythm. This therefore dissociates the putative ultradian rhythm control from the state of being asleep, indicating that sleep and REM–SW alternation are mechanistically independent, at least in part. In sleeping animals, the rhythm could be reset unilaterally, demonstrating the existence of unilateral ultradian oscillators that are normally coupled and in sync. The spontaneous re-syncing of both sides after unilateral reset indicates that the net functional coupling between them is positive. Because bilateral re-syncing after monocular reset could take several sleep cycles, the coupling between left and right CPGs must be weaker than the phase-resetting effect of light on either CPG alone. Finally, the phase-dependent action of a light pulse identifies the middle of SW as a critical time when the effect of an external input switches from phase delay to advance. This result is compatible with considering the first half of SW as a (relative) $REM_P$-refractory period. (In a related but much slower domain, the circadian rhythm shows a qualitatively similar sensitivity to resetting by light: light delivered early during the internal night causes a phase delay; light delivered late causes a phase advance and light delivered any time during the internal day has limited effects[43]). Contrary to SW, $REM_P$ could typically not be stretched beyond its natural duration, suggesting an internally regulated switch to SW. (Whether reset and entrainment of the sleep rhythm can be obtained with other sensory modalities is thus far unknown).

Our results provide a simple functional framework for ultradian sleep rhythm production in this animal; they suggest that mirror-symmetric (L and R) biphasic oscillator circuits drive the alternating production of SW and $REM_P$. Because the features used to characterize SW sleep in *Pogona*, the claustral SWRs, are produced independently in each claustrum[8,9], and because the characteristic claustral features of $REM_P$ (the SNs) originate, by contrast, at or upstream of the isthmus[9], a simple interpretation of our results is that a putative *Pogona* ultradian CPG is located upstream of the isthmus, and that it has two alternating output modes: (1) one identified as 'REM' and characterized by SNs and the absence of claustral SWRs[8,9]; (2) the other identified as 'SW', characterized by the absence of SNs and the enabled the production of SWRs in the claustrum. In this simplest view, $REM_P$ is the relevant output of the CPG and SW is the default state of the forebrain during sleep when $REM_P$ is absent[8]. This view is consistent with the tight regulation we observed for $REM_P$ duration.

The implied existence of paired coupled sleep oscillators upstream of the isthmus is also consistent with the fact that the known pontine CPGs (for example, respiration[31],) are paired, and typically connected via excitatory coupling (for example, by Dbx1-derived commissural neurons[41,44–46]). This is in contrast with spinal CPGs, where bilateral inhibitory coupling usually dominates[11,47]. Our results thus align with what is known about the location and coupling of rhombencephalic CPGs, but are counterintuitive because sleep, characterized by motor inactivity, would then seem to depend on a class of circuits that typically control action. Our proposal, however, finds support in recent work indicating that the neural circuits controlling REM and SW are part of central somatic and autonomic motor systems[48]. A CPG-based perspective on ultradian rhythm control is thus potentially important if these circuits share homologies with those of other pontine CPGs. If so, applying the results of developmental[32,41,45] and transcriptomic studies[49] of motor CPGs (for example, marker genes and rules of circuit architecture), could help decipher, at least in part, the ancestral logic of sleep-control circuits. A CPG-based perspective is helpful also because it could enable both a generic understanding of sleep control, and testable predictions to be formulated. For example, the phase dependency of reset, the different flexibility of $REM_P$ and SW to compression and dilation, the properties of progressive entry into and exit from regular rhythmicity at the beginning and end of the night and the features of bilateral coupling provide clues about these circuits' mechanistic logic as well as constraints for future modelling efforts.

Whereas many investigations of the ultradian sleep rhythm proposed the existence of mutually inhibitory synaptic projections (for example, 'flip-flop'[4,37]), they did not explicitly link them to CPGs. We suspect that this is due to the long periods of the sleep cycle in most studied vertebrates (usually much longer than those of known CPGs), and to the great variety of sleep cycling features, such as regularity and duty cycle. In this respect, *Pogona* sleep is very useful but unusual. Hence, are our results compatible with the great polymorphism of sleep patterns observed across amniotes? The tuning diversity of individual cellular or synaptic properties, the complexification of circuit designs[11,28,29,35–37,50–52] and the addition of internal inputs to the sleep pattern generators as brains evolved could, conceivably at least, result in large numbers of additional control parameters and, thus, in a variety of states and corresponding sleep dynamics. Alternatively, the reptilian rhythm might correspond to a recently identified mammalian noradrenergic infraslow rhythm in the LC that partitions SW sleep into two states of low and high arousability[53–56], defining periodic epochs of potential entry into REM sleep. By this hypothesis, the reptilian rhythm would have been co-opted in mammalian evolution and become a periodic gate that defines when transitions into REM can, but do not necessarily, happen. This in turn would have enabled the duration of each mammalian sleep phase to be under additional levels of control (for example, REM 'pressure'). Note that the period of the noradrenergic infraslow rhythm in mice is shorter (30–50 s) than that of *Pogona*'s SW–REM rhythm (around 130 s at approximately 21.5 °C), but the difference could, in principle, be accounted for by the differences in body temperature and the known temperature dependence of the sleep rhythm frequency in *Pogona*[7]. This will need to be tested. Finally, the properties of motor CPGs are generally influenced by the physical properties of the plants that they move, so that CPG and plant behave coherently as coupled oscillators[40]. If sleep CPGs do exist, their 'plant' must be the brain itself, which, although it does not move, is endowed with its own complex (neural) dynamics. In any species, the properties of sleep would then depend on these interactions and dynamics and thus presumably, on brain complexity. In this regard, the relative simplicity and small size of *Pogona*'s brain may help explain the unusual features of its sleep dynamics.

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

## Methods

### Animals

Adult lizards (*P. vitticeps*) of either sex, weighing 150–300 g (around 1.5–3 years old), were obtained from our institute colony and selected for size, weight and health status. The lizards were housed in our state-of-the-art animal facility. All experimental procedures were approved by the relevant animal welfare authority (Regierungspräsidium Darmstadt, Germany) and conducted following the strict federal guidelines for the use and care of laboratory animals (permit nos. V54-19c20/15-F126/1005_1011 and F126/2006).

### Lizard surgery for chronic recordings

Lizard surgery and electrode implantation was performed as described previously[9]. On the day before surgery, lizards were administered analgesics (butorphanol, 0.5 mg kg$^{-1}$ subcutaneously; meloxicam, 0.2 mg kg$^{-1}$ subcutaneously) and antibiotics (marbofloxacin, marbocyl, 2 mg kg$^{-1}$). On the day of surgery, anaesthesia was initiated with 5% isoflurane, and maintained with isoflurane (1–4 vol%) after intubation. Body temperature was maintained at 30 °C using a heating pad and an oesophageal temperature probe. Heart rate was monitored throughout the surgery using a Doppler flow detector. The skin covering the skull was disinfected using 10% povidone-iodine solution and subsequently removed with a scalpel. Cranial windows were cut to reach the claustrum bilaterally, dura and arachnoid were removed with fine forceps and scissors, and the pia overlaying the dorsal cortex was removed carefully to expose the site of electrode implantation. The exposed skull was covered with ultraviolet-hardening glue and stainless steel wires were secured subdurally to serve as reference and ground.

Silicon probes were mounted on Nanodrives (Cambridge Neurotech) and secured to the skull bilaterally using a stereotactic adaptor. The probes were slowly lowered into both claustra or adjacent anterior dorsal ventricular ridges on the day(s) following the surgery and to a final depth of 0.7–1.3 mm.

The brain was covered with sterile saline, followed by Duragel (Cambridge Neurotech) and Vaseline. The lizards were allowed to recover fully from anaesthesia on a heating pad set to 30 °C before being released into their home terraria.

### In vivo electrophysiology

Before surgery, lizards were habituated for two to three nights to a sleep arena, which was itself placed in a 3 × 3 × 3 m$^3$ electromagnetic-shielded experimental room. An infrared light source was placed on top of the arena and remained switched on for the entire duration of the experiment, allowing continuous monitoring of the animals' behaviour and eye movements using infrared cameras. Around an hour before lights off, the lizards were placed into the arena and left to sleep and behave naturally overnight. They were returned to their home terraria the next day, when they received food and water. Experiments were performed at room temperature of around 21.5 °C. Lizards were entrained to a 12 h/12 h light/dark cycle ($t_{off}$ = 18:00/19:00, winter/summer and $t_{on}$ = 06:00/07:00, winter/summer). Light intensity during the artificial day, measured at the eye, was 15.5 lx (same intensity as for the light pulses).

Electrodes were either 32-channel silicon probes (catalogue no. ASSY-116 H7b, CambridgeNeurotech) and A1x32-Poly2-10mm-50s-177-H32_21mm (Neuronexus) or Neuropixel 1.0 probes. Experiments using 32-channel probes were performed using a Cheetah Digital Lynx SX system, and signals were sampled at 32 kHz. Neuropixel data were acquired nominally at 30 kHz using SpikeGLX software.

IronClust with manual curation was used for spike sorting (https://github.com/flatironinstitute/ironclust#readme) with MATLAB (MathWorks) v.R2021a and v.R2018a.

### Electromyographic recordings

To quantify the difference in muscle tone of sleeping animals under light stimulation and awake animals, we estimated the integral of the electromyographic recording (EMG) (integrated EMG (iEMG), Extended Data Fig. 3) recorded in the animal's neck. The EMG was recorded together with the LFP, sampled at 32 kHz using a Cheetah Digital Lynx SX system, and using a break-out wire (PFE-coated stainless steel Type-316, 125 μm bare) from a 32-channel silicon probe (catalogue no. ASSY-116 H7b, CambridgeNeurotech), insulated except at the tip and inserted into the neck muscle. We band-pass filtered (100–225 Hz) the signal, rectified it (absolute value) and calculated its time integral in a sliding window of 20 s (in steps of 1 ms).

### Light-pulse experiments

Ambient light pulses were applied using LED lights (40 lx measured facing the bulb, 15.5 lx measured as from the eyes of the animal) located above the lizards' sleeping arena and controlled via a USB-to-analogue interface sequence control v.2.0 connected to a personal computer. Pulse protocols started 2–3 h into the dark period, when animals were sleeping deeply and REM$_P$/SW periods alternated regularly (Fig. 1).

In all non-entrainment, single-pulse experiments during sleep (Figs. 1, 2 and 5 and Extended Data Fig. 4), light pulses were delivered between hours 2 and 10 of the recording (between around 20:00 and 04:00). We manually inspected and discarded the few instances in which the sleep rhythm preceding the light pulse was particularly irregular, because it affected our capacity to estimate the phase of the pulse. Hence, 9.15% (or 66 out of 721 in all) of the pulses (all durations) were excluded. No additional criteria (time of night, phase of the pulse or length of the cycle) was used for this selection.

### Sleep entrainment

Light pulses of 1 s were applied at IPIs ranging from 80 to 240 s, 10–17 times in a row. The beginning of each pulse train was separated from the end of the preceding one by at least 30 min.

### Awake entrainment

Experiments took place in the late morning or early afternoon, usually between 10:30 and 13:30, and thus several hours after awakening and during continuous exposure to light. Animals were upright and attentive, with an angled mirror facing the left eye and an infrared camera facing the other, such that movements could be monitored for both eyes (Supplementary Video 2). Each experimental session lasted 30–50 min, and consisted of alternating periods of ambient light OFF and ON (as described above), 9–13 times in a row, before the lights were kept OFF until the end of the experiment.

### Monocular stimulation experiments

For monocular stimulation experiments, a black, three-dimensional printed plastic cup was secured to either left or right eye using silicon (Kwik-Sil). Eye cups were attached around 30 min before starting an overnight recording, and removed the following morning. To pool all unilateral cup experiments (Fig. 5), we use the colour red to indicate the side contralateral to the cupped eye (blind claustrum) and the colour blue to indicate the ipsilateral side to the cupped eye (seeing claustrum), independently of whether they correspond to true right or left.

### Beta power and REM$_P$/SW detection

Note that the nomenclature (REM$_P$ and SW) is descriptive, and does not necessarily imply, for lack of knowledge at this point, functional or mechanistic identity with mammalian sleep states.

We extracted the power of the LFP signal in the beta band (12–30 Hz) for both hemispheres independently with a 10 s sliding window (in 1 s steps) using the Welch method[7,9]. We then combined powers from both hemispheres by taking the maximum value of the beta power

at any one time (Fig. 1b, bottom). To pool diverse recordings while accommodating for rare LFP artifacts, we normalized the beta power so that 0 corresponded to its bottom 5th percentile and 1 to its top 95th percentile. All scale bars for beta correspond to this 0–1 range. Because the timing of the light pulses did not always align with the 1 s sampling of the beta power, we resampled beta around the time of the pulse through linear interpolation with a period of 100 ms.

To define periods of $REM_P$ and SW sleep, we took the $log_{10}$ of the combined beta and fitted a Gaussian mixture model to extract the two peaks of the resulting beta distribution. To avoid over-fragmentation resulting from the noisy crossing of this threshold, we ignored detours that left and re-entered the same state for a short duration (less than 15 s). We used the same method to determine $REM_P$-like and SW-like periods in awake experiments. However, awake experiments were much shorter because the animals generally tolerated only one round of light-pulse entrainment per session; the quality of the Gaussian mixture model fitting was thus reduced. Thus we instead selected thresholds of beta (0.2 for 60 s pulses and 0.4 for 90 s pulses) visually and allowed for shorter detours (less than 5 s).

For the summary in Fig. 5e, we used the above method to define SW (represented in grey) and coloured $REM_P$ in blue or red to identify the claustrum that displayed the highest beta power at any one time. The side with highest beta power dominates the other (SNs are larger and occur 20 ms earlier than on the other side[9]).

To calculate the duration of the cycle, we calculated lagged auto-correlations of the combined beta and determined the lag within the range of 1–3 min that resulted in peak correlation.

To determine when the two hemispheres resynchronized (Fig. 5g), we defined periods of $REM_P$ and SW on the combined beta as described above and determined the beginning of the first $REM_P$ episode after the light pulse that followed a long period of SW (≥40 s).

## Auto-correlations

To compute beta auto-correlations (Fig. 1 and Extended Data Figs. 1 and 3) we calculated Pearson's correlation coefficient from our discrete signal (see 'Beta power and $REM_P$/SW detection' section). For lagged auto-correlations in Fig. 1 we used a sliding window of 20 min and 1 min steps. For Fig. 3 we use a 4 min window and 1 s steps. Formally,

$$c_{\beta,\beta_\tau}(t) = \frac{1}{w} \sum_{\tau=-w/2}^{+w/2} \left[ \frac{(\beta(t+\tau) - \mu_{\beta_\tau})}{\sigma_{\beta_\tau}} \times \frac{(\beta(t) - \mu_\beta)}{\sigma_\beta} \right]$$

where $\beta(t)$ is the beta time series, $w$ is the size of the window, $\mu_{\beta_\tau}$ and $\mu_\beta$ are the means of the lagged and non-lagged beta within the window, and $\sigma_{\beta_\tau}$ and $\sigma_\beta$ are the s.d. values within the window. The resulting value always falls in the range [−1, 1].

## SN and SWR detection

We detected SNs and SWRs as described in ref. 9. In brief, for SNs, we low-pass filtered (40 Hz) the LFP and extracted its first and second derivatives. We then detected triplets of peaks corresponding to the start, middle and end of an SN. To remove false positives, we estimated the distribution of the noise and took only those SNs with a low probability ($P < 0.025$) in the noise distribution of amplitude and duration. For SWRs, we low-pass filtered (30 Hz) the LFP and detected negative peaks using the function scipy.find_peaks. We only considered peaks that were at least 500 ms from one another and that occurred within SW periods (see $REM_P$ detection above).

## Phase analyses and phase response curves

We extracted the phase of either combined (Figs. 2 and 3) or unilateral (Fig. 5 and Extended Data Fig. 6) beta-power (see 'Beta power and $REM_P$/SW detection' section) as described here. We first filtered the beta time series in the band 0.00277–0.16666 Hz using a Butterworth filter. This band preserves the range of timescales relevant to the beta

cycle (0.00277 Hz = 6 min⁻¹, 0.16666 Hz = 1 min⁻¹). We then extracted the Hilbert transform, using the scipy.signal.hilbert function in Python. We next extracted the angle from the complex values of the analytic signal. This process would map the trough, peak and following trough of an ideal sine wave to −π, 0 and π, respectively. As the trough and peak of the band-pass filtered beta correspond to the middle of SW and $REM_P$, respectively, we phase-shift the angles by −π/2 to align −π and π to the beginning of $REM_P$. We then normalized the phase by mapping the range (−π, π] to (0, 1]. The resulting time series maps the beginning and end of SW sleep to 0 and 0.5 and the beginning and end of $REM_P$ to 0.5 and 1. Owing to the properties of the Hilbert transform applied to the smoothened (band-passed) signal, this mapping of 0.5 to the transition of low to high beta remains true even when the duty cycle is not 50%.

Phase is a circular variable. We unwrapped this variable sequentially in Figs. 2e and 5b,c and Extended Data Fig. 6b. We used the function numpy.wrap in Python to detect large deltas that jump from 1 to 0 and added +1 from that point on. If applied to an ideal sine wave with no phase changes, the resulting series moves in a perfect monotonically increasing diagonal (similar to the averages in Fig. 5c and Extended Data Fig. 6b).

For the phase response curves (insets in Figs. 2e and 5b,c and Extended Data Fig. 6b), we calculated the difference of actual phase minus expected phase:

$$\Delta\varphi = \varphi_1 - (\varphi_0 + 1),$$

where $\varphi_1$ is the unrolled phase taken one natural period after the stimulus, and $\varphi_0$ is the phase at the time of the stimulus. The expected phase is the same as the phase of the pulse +1 due to the unwrapping. We obtained the duration of a natural period as described in 'Beta power and $REM_P$/SW detection' and took the actual phase value ($\varphi_1$) from the unwrapped phase series. Because the Hilbert transform involves an integral from both directions in time, our estimation of the phase at which a light pulse fell ($\varphi_0$) was influenced by the outcome of that pulse. To mitigate this effect, we always took the phase 5 s before the actual time of the pulse.

In Fig. 2d–f and Fig. 5b–e, we define the phase ($\varphi$) of the pulse as 'early SW' (blue) if $\varphi \in [0,0.25)$, 'late SW' (green) if, $\varphi \in [0.25,0.5)$ 'early $REM_P$' (orange) if $\varphi \in [0.5,0.75)$, and 'late $REM_P$' (red) if $\varphi \in [0.75, 1)$, where $\varphi = 0$ is the onset of SW.

## Tract tracing of retinal projections

The lizards were anaesthetized as described above (in vivo electrophysiology). Scales covering the skin above the eyeballs were removed carefully, and neurobiotin (20% dissolved in phosphate buffer) was injected intravitreally through a small incision using glass micropipettes, at a rate of 80–120 nl min⁻¹. Lizards recovered from anaesthesia on a heating pad and were subsequently returned to their home terraria. Ten days later, the animals were deeply anaesthetized with ketamine (60 mg kg⁻¹), midazolam (2 mg kg⁻¹) and isoflurane. After loss of the corneal reflex, the lizards were decapitated and their heads perfused with ice-cooled paraformaldehyde (4% in PBS). The brains were extracted and post-fixed with 4% paraformaldehyde–PBS for 24–48 h, and subsequently immersed in 30% sucrose for at least 48 h at 4 °C. Transverse sections were obtained at a thickness of 70 μm, using a cryostat, and neurobiotin was detected with streptavidin, Alexa Fluor 568.

## Statistics

The data were processed using Python (v.3.11.5) and the standard Python packages numpy (v.1.24.3) and xarray (v.2023.6.0). Statistical tests were performed using the standard Python packages scipy (v.1.11.4) and pandas (v.2.0.3).

For the paired two-sided $t$-tests in Fig. 3e,f, we compared the median duration of $REM_P$ (or SW) during the train of light pulses with their median duration in the preceding 25 min. For the Wilcoxon signed-rank paired two-sided tests and Mann–Whitney two-sided $U$-tests of

Extended Data Fig. 3, we took the mean value of the iEMG (see 'Electromyographic recordings' section) in data segments of 6 min.

### Detection of eye opening

To quantify eye openings in our awake experiments (Fig. 4f,g) and during sleep (Extended Data Fig. 3b), we used DeepLabCut (https://github.com/DeepLabCut) to track four points of each eye (midpoint of upper eyelid, midpoint of lower eyelid, left corner and right corner) from our continuous infrared camera recordings. We calculated the Euclidean distance in pixels between the upper and lower eyelids and then normalized them to their 5th and 95th quantiles. Note that, because our video recordings extended throughout each experiment, we tracked the eyes also before sleep started and after it ended, such that a value of 2 or more corresponded to a clear eye opening in awake animals. For Fig. 4g, we considered the eye to be open if this distance was above 0.25.

### Data reporting

No statistical methods were used to predetermine sample size. The experiments were not randomized and the investigators were not blinded to allocation during experiments and outcome assessment.

### Reporting summary

Further information on research design is available in the Nature Portfolio Reporting Summary linked to this article.

## Data availability

All data will be made available upon request.

## Code availability

The code for our analyses is available at: https://brain.mpg.de/research/laurent-department/software-techniques.

57. Hatori, S. et al. Sleep homeostasis in lizards and the role of cortex. Preprint at *bioRxiv* https://doi.org/10.1101/2024.07.31.605950 (2024).

**Acknowledgements** We thank E. Northrup, N. Vogt and S. Dizdarevic for veterinary care; T. Klappich and M. de Vries for reptile care; A. Arends, M. Klinkmann, S. Candlish, J. Knop and C. Thum for technical assistance; F. Kretschmer for help in setting up the automatic light-pulse control; N. Hein for DeepLabCut model training; L. Faraggiana, O. Fernandez, N. Hein and H. Norimoto for discussions; and M. Elmaleh, D. Evans and T. Tomita for their helpful comments on the manuscript. The work was funded by the Max Planck Society (G.L.), the European Research Council (ERC) under the European Union's Horizon 2020 research and innovation programme (grant agreement no. 834446) (G.L.) and the DFG (CRC1080) (G.L.).

**Author contributions** L.A.F., J.L.R. and G.L. designed the project. L.A.F. conceived the experimental designs and carried out the experiments. All authors discussed and interpreted the results. J.L.R. and L.A.F. analysed the data and prepared the figures. G.L. wrote the manuscript, with contributions from J.L.R. and L.A.F., and supervised the project.

**Funding** Open access funding provided by Max Planck Society.

**Competing interests** The authors declare no competing interests.

**Additional information**
**Correspondence and requests for materials** should be addressed to Lorenz A. Fenk or Gilles Laurent.

**a**

**Animal I**

Time of day
— *beta* power CLA-L  — *beta* power CLA-R

6:18 pm
9:18 pm
12:18 am
3:18 am
6:18 am
9:18 am
12:18 pm
3:18 pm

**Animal II**

4:49 pm
7:49 pm
10:49 pm
1:49 am
4:49 am
7:49 am
10:49 am
1:49 pm

**Animal III**

5:07 pm
8:07 pm
11:07 pm
2:07 am
5:07 am
8:07 am
11:07 am
2:07 pm

1 (a.u.)

10 min

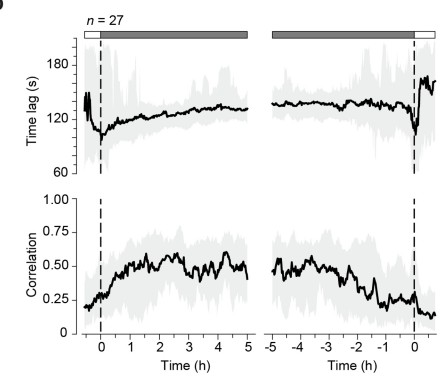

**b**

*n* = 27

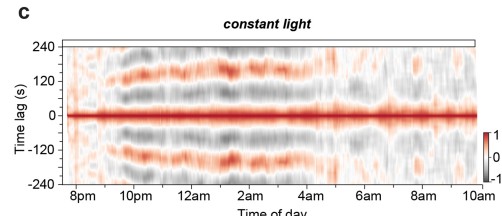
**c**

*constant light*

**Extended Data Fig. 1** | See next page for caption.

**Extended Data Fig. 1 | Further characterization of the ultradian rhythm in *Pogona*. a**, Three 24h-recordings from three different animals (I-III). Each line plots the power of the LFP recorded in the left (blue) and right (red) claustra, in the 12–30 Hz band. For each animal and day, the recordings and plots run from line to line without interruption. Each line represents 3 h. Grey shading: night time (-7 pm to 7am). The insets at right represent the autocorrelogram of the merged (max values, see methods) beta power (time runs down along *y*, time lag in *x*), demonstrating the ultradian rhythm with a period of about 120 s. **b**, Statistics of the ultradian rhythm over 27 nights. The two vertical stippled lines represent the times at which the ambient lights went on and off and serve to align all the recordings. The 2 central hours are clipped to emphasize entry into and exit from the sleep state. Top trace: time lag of peak correlation, corresponding to the period of the ultradian rhythm. This shows that the period typically increases slightly over each night. Bottom trace: shows that the periodicity stabilizes about an hour after dark, and decreases slowly over the last 3 h of the night. Median and 5th to 95th percentile (shading). **c**, Autocorrelogram as in **a** (rotated by 90 degrees) from 8 pm to 10am the next day, showing the ultradian rhythm in an animal held overnight in constant light. This animal is entrained to a normal 12 h light – 12 h dark circadian rhythm and is kept in constant light only during the night of the recording. Note that this animal entered the normal rhythm about an hour later than is typical in darkness, but once asleep, displays the characteristics of the normal ultradian rhythm (about 120 s period, increasing slightly overnight, end before predicted or entrained light-on time).

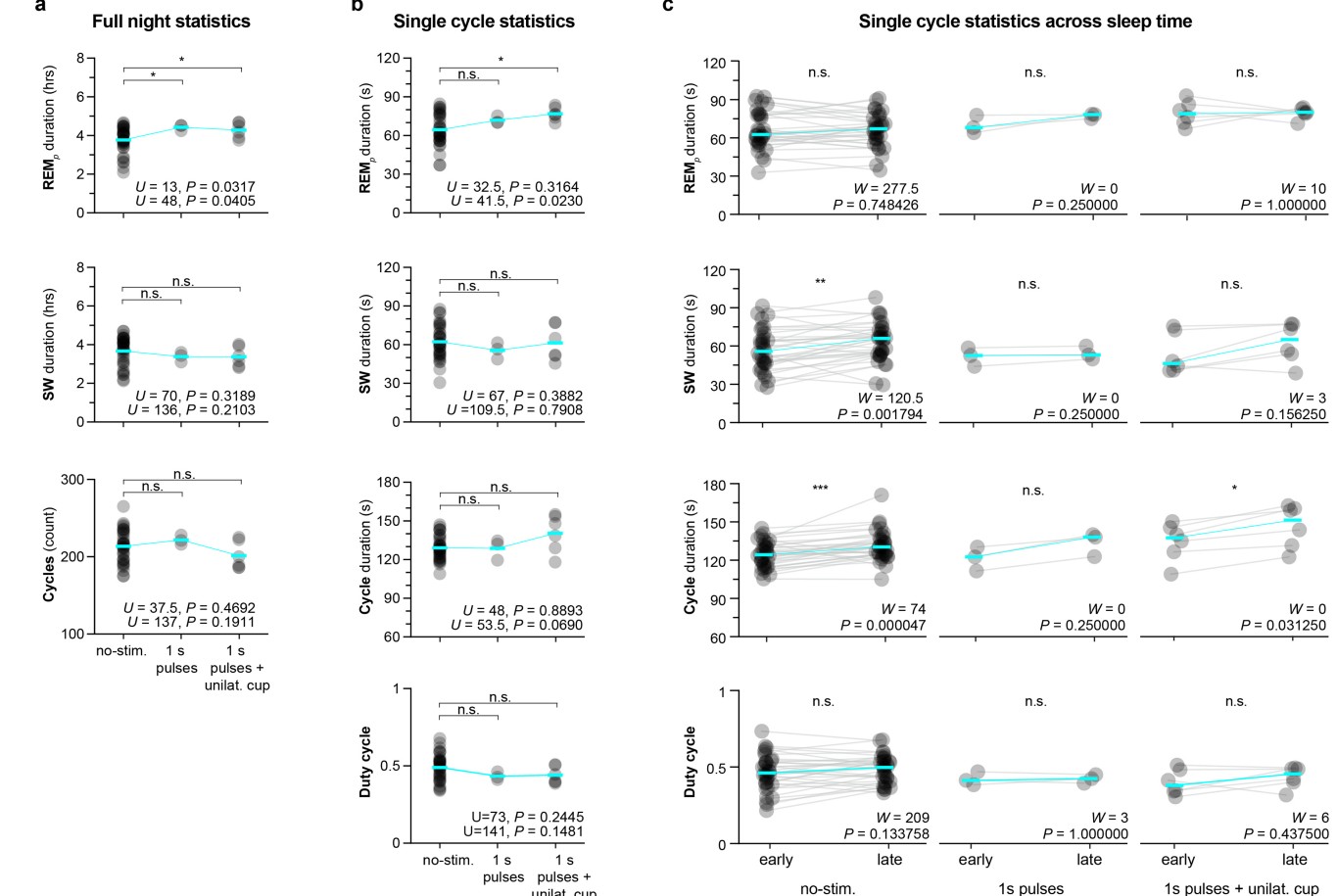

**Extended Data Fig. 2 | Sleep-cycle statistics.** Statistics are calculated from the core 8 h of sleep (~8 pm to 4 am) across 23 animals and 43 nights. **a**, Full night statistics. Top and middle: Total amount of time in $REM_p$ and SW sleep. Bottom: Total number of cycles. We observed a small increase in total REM during experimental nights with light pulses (typically 11 or 12 pulses per night), with values remaining within the range of non-stimulated sleep. Mann–Whitney two-sided U-tests. **b**, Single-cycle statistics. From top to bottom: median duration of a single $REM_p$ episode; median duration of a single SW episode; median duration of the full cycle, calculated as one SW and consecutive $REM_p$;

duty-cycle calculated as the percent of time spent in SW per cycle. We observed a small but significant increase in REM duration in the unilateral cup experiments, within the normal range of non-stimulated sleep. Mann–Whitney two-sided U-tests. **c**, Single-cycle statistics across sleep time. Same as **b** but calculated for the first and last two hours of the core 8 h of sleep. We observed a slight increase of the cycle duration during the night, in the order of 10 s, as previously reported[7]. n = 34 (no-stim), n = 3 (1 s pulses), and n = 6 (1 s pulses + unilat. cup) experiments for each condition. Wilcoxon signed-rank two-sided paired tests.

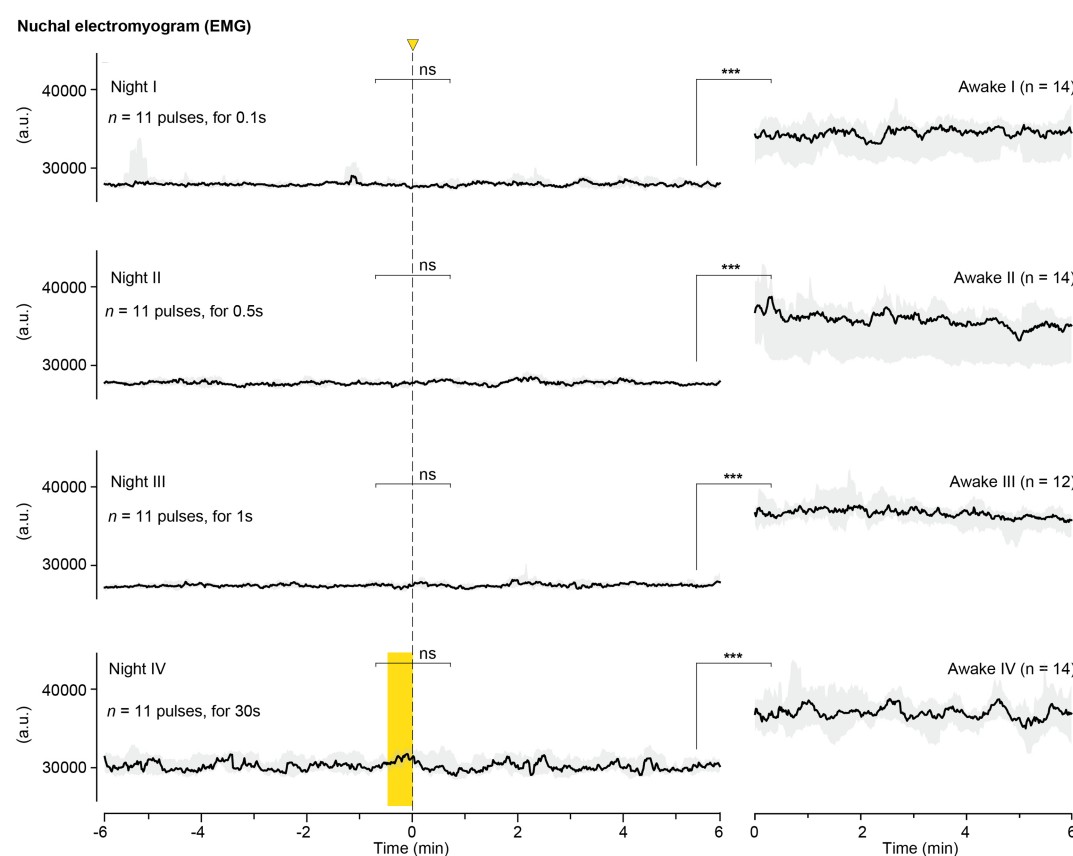

**a** Nuchal electromyogram (EMG)

Night I — *n* = 11 pulses, for 0.1s — ns — *** — Awake I (n = 14)

Night II — *n* = 11 pulses, for 0.5s — ns — *** — Awake II (n = 14)

Night III — *n* = 11 pulses, for 1s — ns — *** — Awake III (n = 12)

Night IV — *n* = 11 pulses, for 30s — ns — *** — Awake IV (n = 14)

Time (min)

**b**

Eye-lid distance

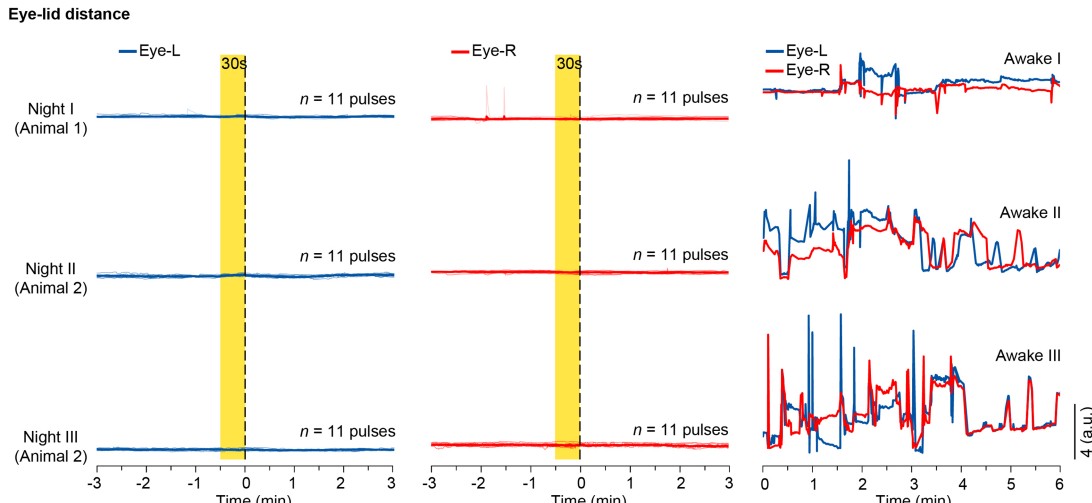

Eye-L — 30s — Night I (Animal 1) — *n* = 11 pulses

Eye-R — 30s — Night II (Animal 2) — *n* = 11 pulses

Night III (Animal 2) — *n* = 11 pulses

Eye-L — Eye-R — Awake I, Awake II, Awake III

Time (min)

4 (a.u.)

**Extended Data Fig. 3 | Ambient light pulses cause neither nuchal electromyographic (EMG) activity nor eye movement. a**, Twelve-min long excerpts of rectified and integrated EMGs (iEMG, see methods) from neck muscles, centered on the end of 0.1 s, 0.5 s, 1 s and 30s-long light pulses (n = 11 in all cases; shown are 4 different nights from one animal). Note the absence of a response to light during sleep, and compare sleep EMG to that recorded in the same animal when it is awake. ***: P < 0.01. Wilcoxon signed-rank two-sided paired test of mean iEMG before and after the light pulse. From top to bottom:

W = 26, P = 0.57715; W = 21, P = 0.32031; W = 18, P = 0.20605; W = 31, P = 0.89844. Mann–Whitney two-sided U-test of mean iEMG after the pulse during sleep and waking state. From top to bottom: U = 4, P = 0.00007; U = 8, P = 0.00018; U = 1, P = 0.00007; U = 0, P = 0.00003. **b**, Measurement of eyelid movements (left and right eyes, see methods) in response to 30 s light pulses in sleeping animals. Note absence of motion, and compare with eyelid movements in awake animals (right). Same calibration in all.

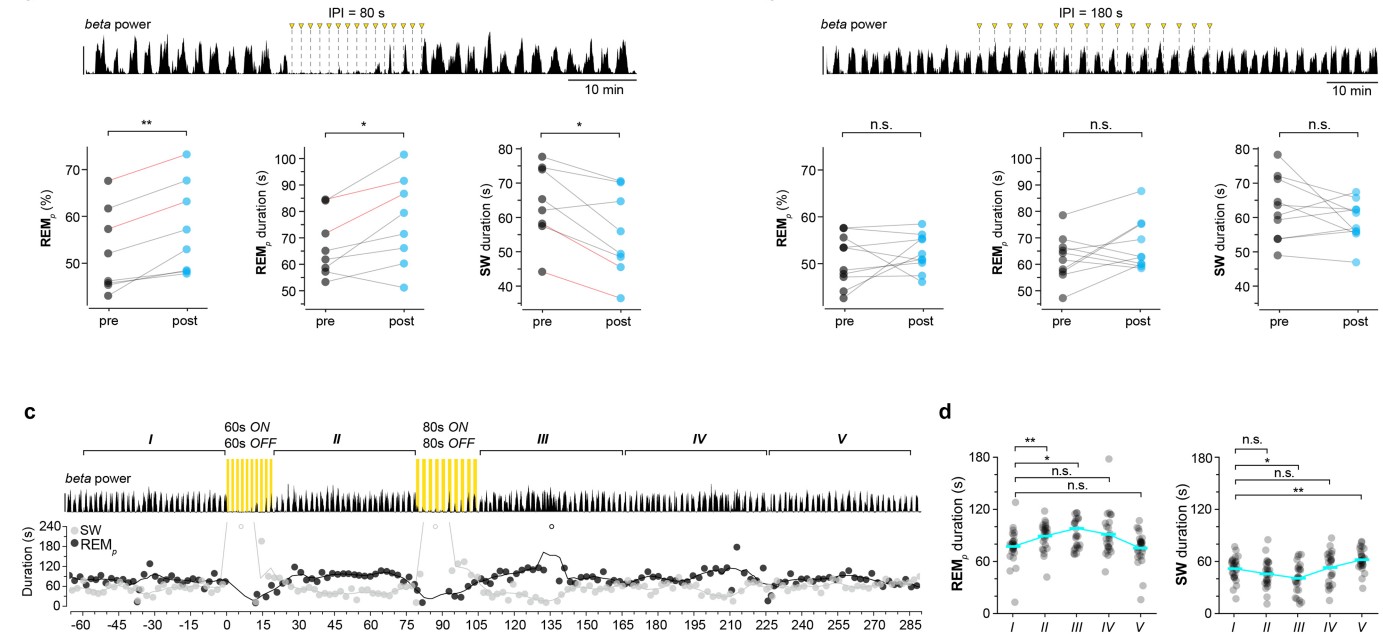

**Extended Data Fig. 4 | REM$_P$ sleep homeostasis. a**, Train of 1s-long light pulses delivered at a short inter-pulse interval (IPI) of 80 s. Such a short IPI fails to entrain the ultradian rhythm, and suppresses REM$_P$ for several minutes (top). Upon cessation of the stimuli, REM$_P$ resumes and occupies a larger fraction of the sleep cycle than before stimulation (bottom left, **P = 0.0078, W = 0), with longer REM$_P$ average duration (bottom middle, *P = 0.0234, W = 2) and shorter SW average duration (bottom right, *P = 0.0156, W = 1); $n$ = 8 experiments; 25 min preceding (pre) and following (post) pulse trains were used for statistical comparison. Wilcoxon signed-rank two-sided paired tests. **: $P$ < 0.05; *: $P$ < 0.1. These results are consistent with a recent study reporting sleep homeostasis in *Pogona*[57]. **b**, Same as in **a** but with light pulses applied at IPI = 180 s, causing reliable entrainment (see Fig. 3). This regime is accompanied by no alteration of the percentage of post-stimulation REM (bottom left, $P$ = 0.375, W = 18), average REM$_P$ duration (bottom middle, $P$ = 0.1602, W = 13) or average SW duration

(bottom right, $P$ = 0.4316, W = 19); n = 10 experiments. **c**, Two consecutive trains of 60 s and 80s-long light pulses, separated by 60 min, and each consisting of 10 pulses delivered every 60 s and 80 s, respectively. Both trains are included in the quantifications of **a** and indicated by red lines. Suppression of REM$_P$ during the light pulses is followed by a rebound, visible as an increase in REM$_P$ and simultaneous decrease in SW duration, slowly returning to baseline levels after stimulation (bottom panel). Open circles indicate long (>240 s) SW or REM$_P$ periods. **d**, Quantification of the data in **c**, comparing the hour preceding the first pulse train (I) with the hour following it (II), and the three hours following the second train (III-V). Left panel (from left to right): **P = 0.0060, U = 196; *P = 0.0225, U = 212.5; $P$ = 0.0555, U = 232.5; $P$ = 0.9361, U = 356. Right panel (from left to right): $P$ = 0.0891, U = 447; *P = 0.0185, U = 466.5; $P$ = 0.5270, U = 290; **P = 0.0066, U = 198. Mann–Whitney two-sided U-tests. ** and * as in **a**.

**a**

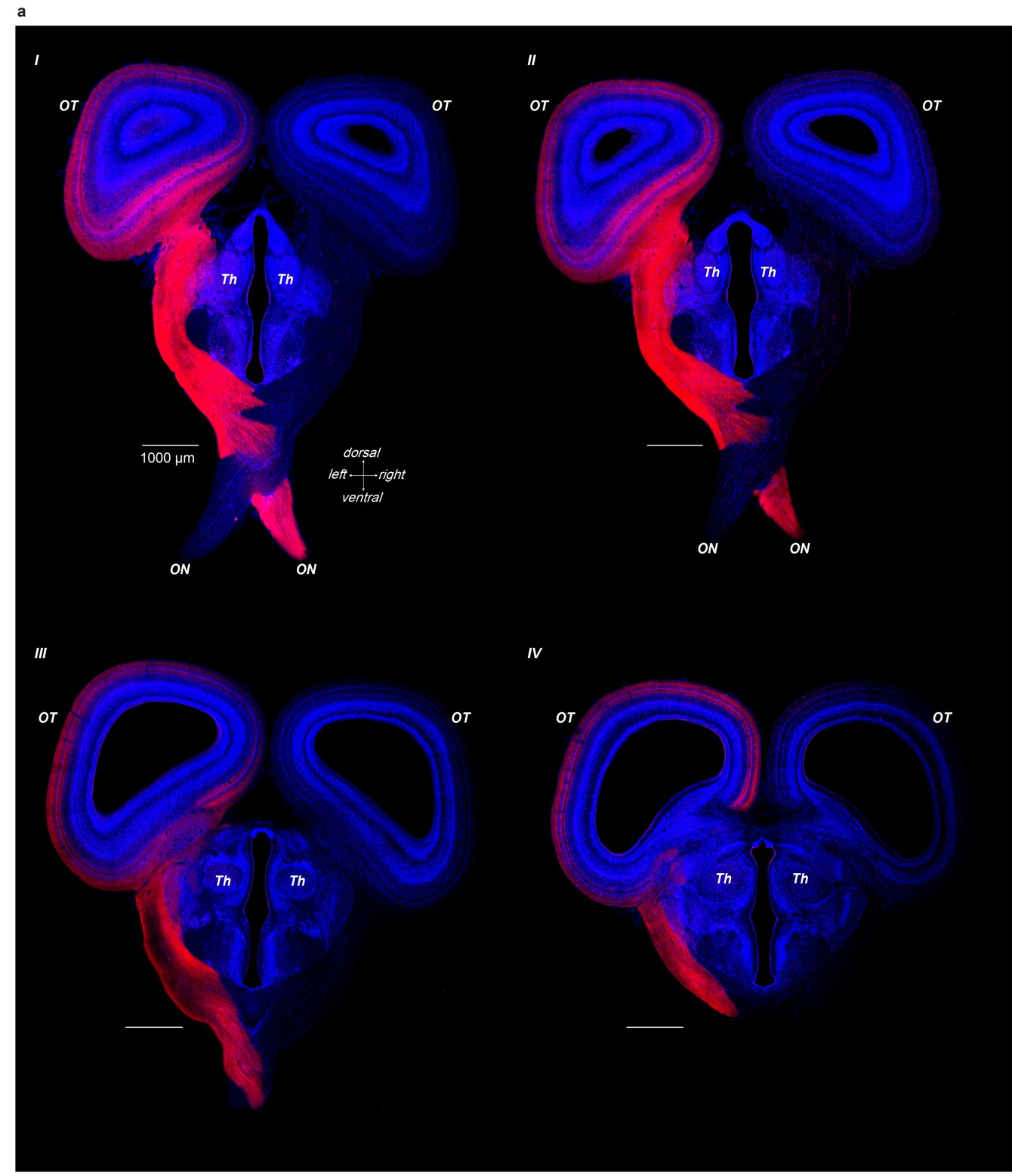

**Extended Data Fig. 5 | Pogona's retinal projections decussate fully at the optic chiasm, enabling monocular reset experiments. a**, I-IV Transverse sections through the brain at the level of the thalamus (Th) and optic tectum (OT) after intravitreal injection of neurobiotin (red) into the right eye. The contralateral labeling suggests the complete decussation of the retinal ganglion cell axons in the optic nerve (ON). Blue = fluorescent Nissl stain.

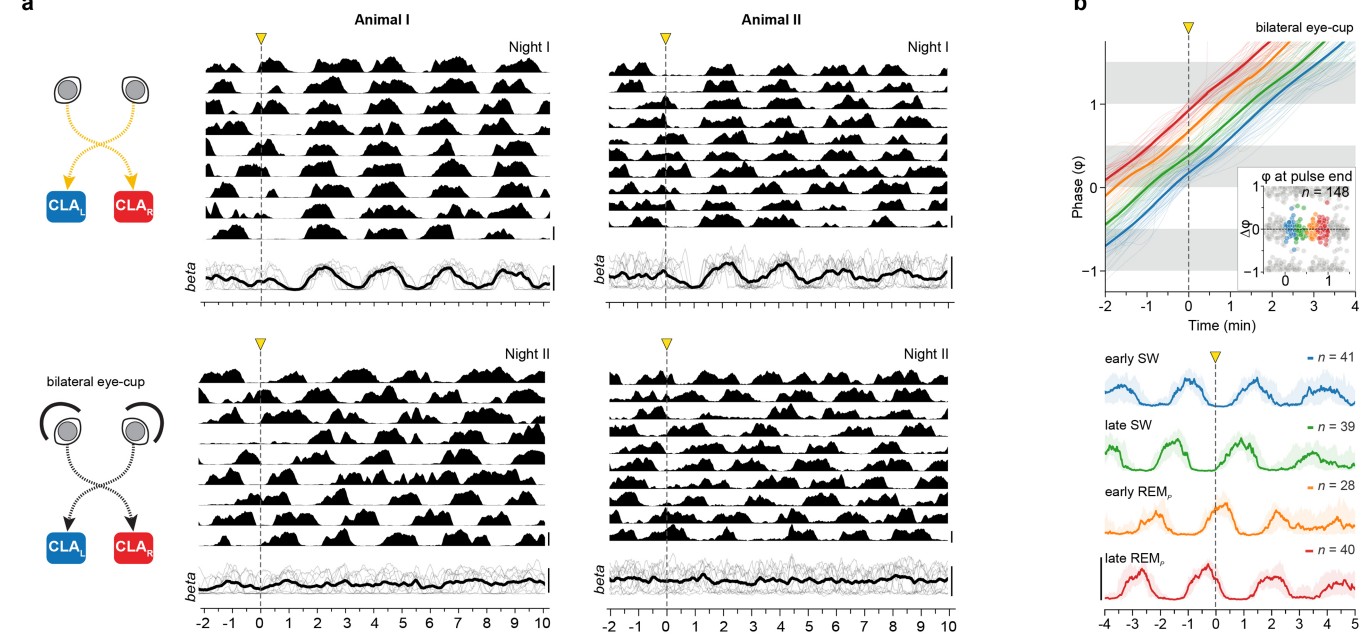

**Extended Data Fig. 6 | Ambient light pulses fail to generate a reset of the ultradian rhythm when both eyes are cupped, proving that the reset by light pulses is due to retinal stimulation through closed eyelids.**
**a**, Combined (L and R) beta power recorded over two nights in two sleeping animals. Top row shows, for each animal and over multiple trials, the light-evoked (1 s long pulses, triangles) reset when no eye cups are present; the bottom row shows the same when both eyes are cupped. Below each panel: superimposed single-trials beta power (grey) and their average (black).

Note that the reset is absent in animals with bilateral eye-cups. **b**, Unrolled phase of claustrum beta in animals with bilateral eye-cups. Calculated from unilateral beta power, as in Fig. 5b,c. Thick (means) and thin (single trials) grouped by phase of light pulse (blue, early SW; green, late SW; orange and red, REM$_P$). Note that responses match those of the blind claustrum in unilateral eye-cup experiments (Fig. 5c). Inset: phase-response curve is flat, indicating no phase-dependent response to the pulse. Below median and IQR of unilateral beta power grouped by phase of the light pulse.

**Extended Data Table 1 | Statistics for entrainment tests in Fig. 3**

| Measure | t-values | p-values | n |
|---|---|---|---|
| **Median REM$_P$ duration (Fig. 3e)** | 3.179, 3.365, 4.190, 0.238, -0.192, -2.647, -1.300, 0.132, 0.947 | 0.024560, 0.028168, 0.002340, 0.821436, 0.855099, 0.026620, 0.263530, 0.901425, 0.397228 | 6, 5, 10, 6, 6, 10, 5, 5, 5 |
| **Median SW duration (Fig. 3f)** | -1.848, -1.422, -1.896, -3.878, -9.564, -11.802, -6.146, -5.494, -0.478 | 0.123837, 0.228154, 0.090462, 0.011669, 0.000212, 0.000001, 0.003555, 0.005347, 0.657856 | 6, 5, 10, 6, 6, 10, 5, 5, 5 |

Two-sided paired t-tests. Values ordered from shortest to longest IPI. n indicates the number of trains of pulses for a given IPI. Data pooled across 12 nights from 4 animals.

Gilles Laurent

# Reporting Summary

## Statistics

For all statistical analyses, confirm that the following items are present in the figure legend, table legend, main text, or Methods section.

| n/a | Confirmed | |
|---|---|---|
| ☐ | ☒ | The exact sample size (*n*) for each experimental group/condition, given as a discrete number and unit of measurement |
| ☐ | ☒ | A statement on whether measurements were taken from distinct samples or whether the same sample was measured repeatedly |
| ☐ | ☒ | The statistical test(s) used AND whether they are one- or two-sided *Only common tests should be described solely by name; describe more complex techniques in the Methods section.* |
| ☒ | ☐ | A description of all covariates tested |
| ☒ | ☐ | A description of any assumptions or corrections, such as tests of normality and adjustment for multiple comparisons |
| ☐ | ☒ | A full description of the statistical parameters including central tendency (e.g. means) or other basic estimates (e.g. regression coefficient) AND variation (e.g. standard deviation) or associated estimates of uncertainty (e.g. confidence intervals) |
| ☐ | ☒ | For null hypothesis testing, the test statistic (e.g. *F*, *t*, *r*) with confidence intervals, effect sizes, degrees of freedom and *P* value noted *Give P values as exact values whenever suitable.* |
| ☒ | ☐ | For Bayesian analysis, information on the choice of priors and Markov chain Monte Carlo settings |
| ☒ | ☐ | For hierarchical and complex designs, identification of the appropriate level for tests and full reporting of outcomes |
| ☐ | ☒ | Estimates of effect sizes (e.g. Cohen's *d*, Pearson's *r*), indicating how they were calculated |

*Our web collection on statistics for biologists contains articles on many of the points above.*

## Software and code

Policy information about availability of computer code

| Data collection | Cheetah (Neuralynx) for recordings using 32-channel CambridgeNeurotech and NeuroNexus probes (version 6.4.2). SpikeGLX for Neuropixels recordings (v.20200309);. Zen 2.1 and 3.1 (Carl Zeiss) was used for image acquisition. |
|---|---|
| Data analysis | Python (version 3.11.5) and MATLAB (MathWorks) version R2021a and R2018a. Python packages used: scipy (1.11.4), numpy (1.24.3), pandas (2.0.3), and xarray (2023.6.0). Code available at: https://brain.mpg.de/research/laurent-department/software-techniques DeepLabCut (version 2.3.0). Ironclust (4.2.3) was used for spike sorting and manual curation. |

For manuscripts utilizing custom algorithms or software that are central to the research but not yet described in published literature, software must be made available to editors and reviewers. We strongly encourage code deposition in a community repository (e.g. GitHub). See the Nature Portfolio guidelines for submitting code & software for further information.

## Data

Policy information about availability of data

All manuscripts must include a data availability statement. This statement should provide the following information, where applicable:
- Accession codes, unique identifiers, or web links for publicly available datasets
- A description of any restrictions on data availability
- For clinical datasets or third party data, please ensure that the statement adheres to our policy

> Data will be available upon reasonable request.

## Research involving human participants, their data, or biological material

Policy information about studies with human participants or human data. See also policy information about sex, gender (identity/presentation), and sexual orientation and race, ethnicity and racism.

| | |
|---|---|
| Reporting on sex and gender | not applicable |
| Reporting on race, ethnicity, or other socially relevant groupings | not applicable |
| Population characteristics | not applicable |
| Recruitment | not applicable |
| Ethics oversight | not applicable |

Note that full information on the approval of the study protocol must also be provided in the manuscript.

# Field-specific reporting

Please select the one below that is the best fit for your research. If you are not sure, read the appropriate sections before making your selection.

☒ Life sciences ☐ Behavioural & social sciences ☐ Ecological, evolutionary & environmental sciences

For a reference copy of the document with all sections, see nature.com/documents/nr-reporting-summary-flat.pdf

# Life sciences study design

All studies must disclose on these points even when the disclosure is negative.

| | |
|---|---|
| Sample size | No statistical tests were used to predetermine sample sizes. We established that our sample sizes are sufficient based on previous experience and commonly used sample sizes in this field of research, taking into account the unusual nature and limited availability of the animal species studied. |
| Data exclusions | Experiments with off-target placement of electrodes, or tracer injections were excluded from our analysis. |
| Replication | Main results contain multiple data sets in which findings could be replicated (see exact n-numbers in Figures and their legends) |
| Randomization | Animals were not assigned to groups, and were selected based on weight and healthy appearance. Randomization was not relevant for our study. |
| Blinding | Investigators were not blinded to group allocation during data collection and analysis. Our study was mostly observational in nature, measurements were fully automated, and blinding was thus not relevant for our study. |

# Reporting for specific materials, systems and methods

We require information from authors about some types of materials, experimental systems and methods used in many studies. Here, indicate whether each material, system or method listed is relevant to your study. If you are not sure if a list item applies to your research, read the appropriate section before selecting a response.

## Materials & experimental systems

| n/a | Involved in the study |
|-----|----------------------|
| ☒ | Antibodies |
| ☒ | Eukaryotic cell lines |
| ☒ | Palaeontology and archaeology |
| ☐ | ☒ Animals and other organisms |
| ☒ | Clinical data |
| ☒ | Dual use research of concern |
| ☒ | Plants |

## Methods

| n/a | Involved in the study |
|-----|----------------------|
| ☒ | ChIP-seq |
| ☒ | Flow cytometry |
| ☒ | MRI-based neuroimaging |

## Animals and other research organisms

Policy information about studies involving animals; ARRIVE guidelines recommended for reporting animal research, and Sex and Gender in Research

| Laboratory animals | Adult Lizards (Pogona vitticeps), weighing 150-350g (age ~1.5 to 3years) bred and housed in our state-of-the-art animal facility. |
|---|---|
| Wild animals | This study didn't involve wild animals. |
| Reporting on sex | Our findings apply to either sex, which has been assigned based on visual and/or ultrasound inspection. Experimental animals have been chosen based on availability, weight and healthy appearance, irrespective of their sex. We did not perform any sex-based analysis, but our results are highly consistent across all animals studied. |
| Field-collected samples | This study did not involve field-collected samples. |
| Ethics oversight | All experimental procedures were approved by the relevant animal welfare authority (Regierungspräsidium Darmstadt, Germany) and conducted following the strict federal guidelines for the use and care of laboratory animals (permit numbers V54-19c20/15-F126/1005_1011 and 2006). |

Note that full information on the approval of the study protocol must also be provided in the manuscript.

## Plants

| Seed stocks | NA |
|---|---|
| Novel plant genotypes | NA |
| Authentication | NA |

