## [Peer Review File · Nature]

Manuscript Title: Central pattern generator control of a vertebrate ultradian sleep rhythm

Reviewer Comments & Author Rebuttals

Reviewer Reports on the Initial Version:

Referees' comments:

Referee #1 (Remarks to the Author):

In the manuscript by Fenk, Riquelme and Laurent, the authors investigate the idea that sleep stage transitions could be regulated by a form of central pattern generator (CPG), similar to that described in motor systems. They use reptiles (*Pogona*) to test whether evidence of CPG can be found during sleep, such as modifications to phase transitions by external stimuli (specifically light stimuli). They found that light stimuli at different frequencies, could trigger a phase shift of the ultradian sleep cycle depending on the part of the NREM cycle that the stimuli were provided. Furthermore, the ultradian rhythm could be modified to be faster or slower by stimuli given within a specific frequency range, consistent with an internal frequency governing this cycle (such as that in CPG that regulate motor systems). Delivering the light stimulus through only one eye, could modify the ultradian rhythm through the contralateral, but not ipsilateral hemisphere, and after the reset was induced, both hemispheres eventually resynchronized. I found the idea of the study novel and compelling. And although the study lacks mechanistic insights that could help understand their findings and link them to existing literature, the results are convincing as an evidence of phase transition modifications, but I suggest that the authors provide further analyses and experiments to fully support their hypothesis. Some comments are below:

-In general, throughout the manuscripts there are missing quantifications specially related to the average duration of sleep stages (REM / NREM) and across sleep time. These quantifications should be provided, for each figure, in the form of a panel (bar or lines over time for each animal...) instead of only a number in the text, as it can provide valuable information and easy comparisons for the reader to follow the argument.

-Do the behavior of the animals differ when stimulus are being given? For example, locomotion patterns, etc.? Please quantify.

-Figure 1. Quantifications (histograms with errors) of these findings should follow the observations. Number of transitions REM/NREM per night and hemisphere as well as across nights. Along the same lines, length of REM/NREM varies across sleep time in mammals, is this the case in *Pogona*? The rate of SWRs vary in mammals across the sleep, being higher at the beginning of the NREM sleep and decreasing with time, is this the same in *Pogona*?

-Figure 1d-f, if this is data from 3 animals, do traces indicate the mean? Please show error bars/shadow. Are scale bars in f applicable to d and e?

-The authors write "Sensory inputs such as light, sound or touch delivered to sleeping animals typically awaken them if sufficiently intense." I believe a reference should be added since authors will make a

strong point later about applying stimulation that is not enough to awake the animals. A related point to this is that, the authors use light as an external stimuli, but do they think that the results will be preserved with other sensory modality? They should provide an experimental test that rule out this possibility, as it will make their claims stronger.

-Figure 1g: is the total amount of REM preserved across nights? Please quantify.

-Figure 2d: please define what is considered “early” and “late” NREM/REM, is it first and second half of a given episode? Please describe in methods.

-Figure 5a: methodology for eye occlusion is incomplete, is unclear how this was done and whether the animal displayed normal sleeping patterns after this manipulation. Please quantify and compare to regular sleeping nights (animals not blinded) the amount of REM/NREM overall, the duration of each cycle and over time, the number of transitions... For example, are firing rates of units changed during these conditions of eye closed? Since the authors have the data, please quantify.

-In general, firing rates results should be included for every experiment (i.e., light stimulation), to get a better understanding of how different parameters regulate individual cells.

-Figure 5g: this result is interesting but not convincing. Please provide further quantifications of this (not only the phase of the stimulus but the timing within the overall sleep for example) and separate or provide quantification for different animals, to rule out the possibility of inter-subject variability. Were all stimuli given for the same duration? Doing experiments where distinct durations of the stimulus are given, could provide insights into how the re-synchronization happen. One way would be to analyze units activity, as the authors state in the methods that they have clustered the data into single neurons using IronClust. If re-synchronization is happening due to an internal mechanism, as the authors suggest, it should be quantifiable with cross-correlograms of neurons from each hemisphere.

-In the methods, the authors say that “the lizards were placed into the arena and left to sleep and behave naturally overnight. They were returned to their home terraria the next day, when they received food and water. Experiments were performed at room temperature of around 21.5 °C.” Does this mean that during the time that data is being analyzed (sleeping in the arena) animals were water and food deprived? Please clarify.

Referee #2 (Remarks to the Author):

This manuscript explores the hypothesis that *Pogona*'s NREM-REM ultradian rhythm is produced by a central pattern generator (CPG). Specifically, the authors test whether the ultradian rhythm can be reset in a phase-dependent manner and whether it exhibits entrainment by light pulses. They also used a short light pulse delivered to a single eye to establish that sleep is generated by paired rhythm-generating circuits linked by functional excitation. This is innovative work in an understudied animal model, and there are interesting features of *Pogona*'s sleep [e.g., a short (~2 min) period and high regularity] that are of general interest to sleep researchers and that motivate the use of this animal model to address these questions. The demonstration that “REM can be shortened but typically not extended beyond a natural limit (p.5)” is elegant and compelling. However, I have some reservations about some of the interpretation/conclusions, and I feel that some aspects of the manuscript require

more context. Below, I have detailed my questions and concerns regarding the manuscript.

Major concerns:

1. The second paragraph of the introduction describes two hypothesized mechanisms for REM sleep generation, and the third paragraph states that both of these mechanisms require “reciprocally inhibitory circuits.” However, the term “reciprocal interaction” described in the first mechanism refers to inhibitory- excitatory coupling between monoaminergic and cholinergic neurons, respectively, in the conceptual and mathematical model proposed by McCarley and Hobson. These two hypothesized mechanisms actually generate cycles using distinct dynamic structures (hysteresis loop vs. limit cycle) as described in ref 27 (Diniz Behn et al., 2013). The description of the reciprocal interaction circuit should be corrected to clarify the difference between the mechanisms.
2. Ref 7 (Shein Idelson et al. , 2016) indicates that the period of the ultradian rhythm in *Pogona* varies over the course of the night as it does in humans. However, the trial-to-trial variability (e.g., in Fig 1g) seems very small. Were additional (time of night?) criteria used to select the cycles included in these analyses?
3. Ref 7 also mentions that the period of the ultradian rhythm is temperature dependent. The issue of temperature dependence in these rhythms should be addressed, at least in the Discussion.
4. In mammalian species, there is evidence for a homeostatic pressure for REM sleep (e.g., Endo et al., AJP, 1998). Is there similar evidence for REM homeostasis in *Pogona*? Specifically, does the alternation between high and low claustral beta show any evidence of modulation by REM homeostasis or change in response to a homeostatic challenge?
5. Relatedly, a correlation between the duration of a REM episode and the duration of the following NREM episode has been reported in mammals including rats, cats, monkeys, and humans (e.g., Vivaldi et al., J Neurophysiol, 1994; Benington and Heller, AJP, 2000; Barbato et al., SLEEP, 1998). Does this association hold in the ultradian cycles observed in *Pogona*?
6. The authors interpret the middle of NREM as a “critical phase of the sleep cycle” that separates phase advancing/delaying effects of light pulse. Can this interpretation be distinguished from the existence of a REM refractory period (cf. LeBon, Sleep Medicine, 2020) during which REM-like behavior cannot be initiated? The presence of a refractory period would be supported by the results in Fig 2f showing the response to a 90s light pulse when, in the top panel, the beta activity is eventually triggered.
7. There has been a recent focus on the role of an infraslow rhythm modulating the microstructure of sleep in rodents (e.g., Stucynski et al., Curr Biol, 2022; Kjaerby et al., Nat Neurosci, 2023). Given the timescale of the ultradian oscillation in *Pogona*, it would be interesting to know if there is any evidence for an infraslow rhythm modulating sleep in this amniote.

Minor concerns:

1. The Introduction should include more background on *Pogona*'s sleep behavior for readers not familiar with previous work (e.g., ref 7). It would be helpful to review the diurnal behavior pattern and occurrence of a consolidated sleep period.
2. The term “reciprocal interaction” is often used to describe the excitatory-inhibitory coupling in the

McCarley-Hobson model. It would be best to avoid this terminology when referring to mutually inhibitory coupling to avoid confusion.

3. The interpretation of high beta power as a marker of REM sleep is not intuitive to researchers familiar with mammalian sleep. Some of the justification for this marker provided in refs 7 and 9 should be included here.

4. In the transitions between periods of high and low beta associated with REM and NREM sleep, respectively, what is the threshold used to distinguish between states, and how is it selected?

5. It sounds like the first sentence on p. 4 (“Because the animals...lights go out.”) is intended to describe anticipatory behavior resulting from entrainment. However, this meaning should be clarified since this interpretation of eyes closing spontaneously is not obvious. This idea was stated more clearly in ref 7.

6. P. 4 The authors describe the oscillator circuit as “activated and inactivated by circadian clock.” This implies both a positive and negative action by the clock at different times/phases. Is this the intention, or could this be regulation better described as “gating by” the circadian clock? If the former, is there evidence for positive and negative action?

7. Fig 3b: how is preferred phase defined?

8. The Methods should specify that the lizards were entrained to a 12:12 Light:Dark cycle with time of lights on and intensity of light during the light period.

9. There was minimal time for habituation to the eye cups used for the monocular simulation experiments. Was there any evidence of stress in the animal and/or abnormal sleep behavior associated with the use of eye cups?

10. In the REM detection experiments, the authors note that “the animals generally tolerated only one round of light pulse entrainment per session.” What does “tolerated” mean in this context? Could the REM-like and NREM-like behavior only be elicited once? What might this mean for the circuit?

11. Phase analyses and phase response curves: Is there a typo in “band pass filtered the beta time series by 0.00277 – 0.16666Hz (corresponding to 1/6 minutes and 1/1 minute)”? Not sure what this is saying.

12. Top of p. 17: Choice to evenly divide interval between NREM and REM seems odd in situations where NREM and REM are not evenly divided across the ultradian cycle. Authors should comment on the appropriateness of this choice.

Referee #3 (Remarks to the Author):

The paper presents convincing evidence that the biphasic sleep pattern observed in the *Pogona* lizard is governed by a CPG-type oscillator process. The experiments and reported results are aligned with the typical manipulations conducted to determine the existence of a CPG such as phase shifting and entrainment. This is a very interesting finding pointing to novel hypotheses about the neural mechanisms controlling sleep and transitions between sleep states. The results provide new insights into processes that may govern and contribute to sleep patterning in other animals, including mammals. The sleep community will be particularly interested in these results but they should also have broader appeal as an interesting example of an evolutionarily simpler process to govern sleep.

I am skeptical that REM and NREM state alternation in mammals is governed by the same mechanism. Particularly given the direct effect of light on the identified CPG process shown here while in mammals

circadian modulation from the SCN to sleep-promoting centers is polysynaptic and not entirely identified. However, the reported CPG-process could be related to mechanisms responsible for recently identified slow (~50s) rhythmic activity in hypothalamic and brainstem areas that appears during NREM sleep. Specifically, recent studies have shown that the EEG during NREM sleep in both humans and mice shows a pronounced infraslow modulation in the sigma (10-15 Hz) power range (see PMC5298853, PMC8752505, PMID: 34648731 for example). Transitions out of NREM sleep to both wake and REM sleep are synchronized with this infraslow sigma power rhythm. This alternation in brain states marked by activity in different frequency bands during sleep on a slow time scale is not entirely dissimilar from the *Pogona* sleep patterning. The mechanisms responsible for the infraslow sigma band rhythm in mammals is not known but may also be neural in origin, as hypothesized for the mechanism of the CPG process reported here. I think it would increase the impact of the results for the authors to include a discussion of a potential relation to this sigma band rhythm; the *Pogona* beta power sleep rhythms may perhaps be an evolutionarily earlier mechanism for rhythmic patterning of different brain activity levels during sleep that shape the well-known NREM-REM alternation.

Major point:

In my opinion, the attempt to relate the alternation of the high and low beta power sleep states in the *Pogona* to NREM-REM alternation in mammals is made a little too strongly. Until further experiments are done that replicate key properties of NREM-REM alternation in mammals, such as response to sleep deprivation, the competition of NREM and REM sleep rebound after deprivation, homeostatic regulation, etc, it's better if the authors don't conflate the two. For example, the 2nd paragraph in the Introduction reviewing hypotheses for mammalian NREM-REM alternation may be better placed in the Discussion so that the relationship between the presented phenomena and mammalian NREM-REM alternation is not immediately presumed. Also, on pg 10, end of middle paragraph, the existence of REM-on and REM-off neurons in mammals should not be related to the putative CPG in the *Pogona*.

In their previous paper Fenk et al Nature 2023, the authors use the notation REMp for the sleep state with high beta activity and explicitly state that the NREM and REM "nomenclature is descriptive, and does not necessarily imply, for lack of knowledge at this point, functional or mechanistic identity with mammalian sleep states." To better avoid the conflation with mammalian NREM-REM alternation, they should likewise use that nomenclature here and make the same statement.

The CPG-type oscillator process responds directly to light and all manipulations of the CPG process are in relation to the action of external light on that process. Without knowing what the CPG process is, it may well be possible that there are other stimuli that can perturb the process and have different phase shifting or resetting effects. The authors should make it clear in statements that interpret the function of the CPG process that the results show only the effects of light pulses on the CPG process. For example, on pg 5, last sentences of the 1st paragraph, the authors state "In other words, REM can be shortened but typically not extended beyond a natural limit, whereas NREM can be both shortened and lengthened." Here, this is shown only for the effect of a light pulse. There may be other stimuli that could lengthen REM episodes if more were known about the CPG process. Similar statements that need this qualification are:

Last sentence on pg 5: clarify that light is the external stimulus.

Pg 10, last sentence of top paragraph: clarify that light could not increase REM duration suggesting that processes governing the switch to NREM are not affected by light.

Minor point:

Fig 3b: Is the phase on the x-axis defined relative to the onset of low or high beta in the spontaneous NREM-REM pattern? Namely, how is phase = 0 defined? More information about the definition of the phase should be added to the figure caption.

Pg 6, bottom paragraph: "...with a widening of the phase distribution away from the optimum" This effect is not strongly visible except for the IPIs when no entrainment occurred. Maybe be more clear about what the reader should be looking for in the figure to support this statement.

Overall, the paper is well-written and clear. The validity of approach and quality of data are excellent. Data analyses and statistical analyses are appropriate and well done.

Author Rebuttals to Initial Comments:

Nature manuscript 2024-05-10635 by Fenk, Riquelme and Laurent

Responses to referees' comments.

We thank the referees for their many and detailed comments, which we address below. All changes to the manuscript are highlighted in the updated ms text file, to which we have added 2 new Extended Data Figures. These changes and all other issues raised are detailed point by point in the response below (in blue).

Referee #1 (Remarks to the Author):

In the manuscript by Fenk, Riquelme and Laurent, the authors investigate the idea that sleep stage transitions could be regulated by a form of central pattern generator (CPG), similar to that described in motor systems. They use reptiles (*Pogona*) to test whether evidence of CPG can be found during sleep, such as modifications to phase transitions by external stimuli (specifically light stimuli). They found that light stimuli at different frequencies, could trigger a phase shift of the ultradian sleep cycle depending on the part of the NREM cycle that the stimuli were provided. Furthermore, the ultradian rhythm could be modified to be faster or slower by stimuli given within a specific frequency range, consistent with an internal frequency governing this cycle (such as that in CPG that regulate motor systems). Delivering the light stimulus through only one eye, could modify the ultradian rhythm through the contralateral, but not ipsilateral hemisphere, and after the reset was induced, both hemispheres eventually resynchronized. I found the idea of the study novel and compelling. And although the study lacks mechanistic insights that could help understand their findings and link them to existing literature, the results are convincing as an evidence of phase transition modifications, but I suggest that the authors provide further analyses and experiments to fully support their hypothesis.

We thank the referee for his/her comments.

Some comments are below:

-In general, throughout the manuscripts there are missing quantifications specially related to the average duration of sleep stages (REM / NREM) and across sleep time. These quantifications should be provided, for each figure, in the form of a panel (bar or lines over time for each animal...) instead of only a number in the text, as it can provide valuable information and easy comparisons for the reader to follow the argument.

Our previous papers (Shein-Idelson et al., 2016; Norimoto, Fenk et al., 2020; Fenk et al, 2023) provided detailed descriptions of the statistics of Pogona sleep, in particular detailing its reliability, regularity, time and temperature dependency etc. We have, however, added the requested data in the present ms., as appropriate and requested, in the form of a new Extended Data Fig. 2 (also below), which provides data on total and median values characterizing sleep at different times of the night, without and with light pulses, and with or without eye occlusion. Most pairwise comparisons reveal no differences, and a few identify small but barely significant differences. We address the reviewer's individual concerns below.

(New) Extended Data Fig. 2 | Sleep-cycle statistics calculated from core 8h of sleep (~ 8pm to 4am) across 23 animals and 43 nights. a, Full night statistics. Top and middle: Total amount of time in REM_p and SW sleep. Bottom: Total number of cycles. We observed a small increase in total REM during experimental nights with light pulses (typically 11 or 12 pulses per night), with values remaining within the range of non-stimulated sleep. Mann–Whitney U-tests. b, Single-cycle statistics. From top to bottom: median duration of a single REM_p episode; median duration of a single SW episode; median duration of the full cycle, calculated as one SW and consecutive REM_p; duty-cycle calculated as the percent of time spent in SW per cycle. We observed a small but significant increase in REM duration in the unilateral cup experiments, within the normal range of non-stimulated sleep. c, Single-cycle statistics across sleep time. Same as b but calculated for the first and last two hours of the core 8h of sleep. We observed a slight increase of the cycle duration during the night, in the order of 10s, as previously reported (Shein-Idelson et al., 2016). $n = 34$ (no-stim), $n = 3$ (1s pulses), and $n = 6$ (1s pulses + unilat. cup) experiments for each condition. Wilcoxon signed-rank paired tests.

-Do the behavior of the animals differ when stimulus are being given? For example, locomotion patterns, etc.? Please quantify.

With the exception of the awake entrainment experiments, all stimuli were provided to sleeping animals, thus not displaying any behavior (locomotor or other). As described in the ms, the light stimuli led to no detectable behavioral effects, as illustrated in Movie 1 and quantified in Extended Data Fig. 3 (EMG and eye-tracking).

During the awake entrainment experiments (Fig 4), the animals had an erect posture, typical of awake Pogona, which they kept as light was turned off and on. Turning lights off led, after several seconds, to the animal closing their eyes. Turning lights back on led, with typically shorter latency, to the animal opening their eyes, as illustrated in Fig. 4g, f, and in movie 2.

-Figure 1. Quantifications (histograms with errors) of these findings should follow the observations. Number of transitions REM/NREM per night and hemisphere as well as across nights.

Quantifications are now provided in a new EDF (Extended Data Fig. 2). The number of transitions REM/SW (Extended Data Fig. 2a, cycle count) was around 200 in the core 8h of sleep analyzed here, with no significant difference between non-stimulated and stimulated nights. Also note: the cycling between SW and REM is synchronized across hemispheres (Fenk et al., 2023), and both sides of the brain go through an equal number of REM/SW transitions.

Along the same lines, length of REM/NREM varies across sleep time in mammals, is this the case in Pogona?

These features were detailed in Shein-Idelson et al., 2016. Although the sleep cycle duration increases slightly during the night (Extended Data Fig. 2c , Cycle duration), the duty cycle remains around 0.5 (Extended Data Fig. 2c , Duty cycle), with SW taking just under 50% of the cycle (see also Shein-Idelson et al., 2016, Fig. S7)

The rate of SWRs vary in mammals across the sleep, being higher at the beginning of the NREM sleep and decreasing with time, is this the same in Pogona?

We report the statistics of SWRs (detected algorithmically) over multiple normal (i.e., non-stimulated) nights (4 animals, 7 nights) in the figure below. We observed no significant change in the rate of SWR within single SW epochs (a) and a slight trend towards fewer SWRs as the night progressed (b). Note that the SWRs analyzed here (and reported in Fig. 4) are produced in the claustrum (Norimoto, Fenk et al., 2020) independently across both hemispheres (Fenk et al., 2023, see Fig. 2b), and that SW epochs in Pogona are short (60-90s, see new Extended Data Fig. 2b, c). These differences may make direct comparisons with mammalian hippocampal SWR statistics difficult.

Response-to-Reviewers Figure 1: Sharp-wave ripple (SWR) rate changes over sleep duration.

Data pooled from both hemispheres across 7 non-stimulated nights in 4 animals. **a**, SWR rate in the first and last 30s of any SW period of at least 90s. Dots and bars indicate mean and std per hemisphere and recording ($n = 14$). Population mean in red (Wilcoxon signed-rank test. $W = 22$, $P = 0.058$). **b**, Top: Rate of SWRs within slow-wave sleep (SWS) epochs in 30-minute-long sliding windows. Mean in red. Bottom: Same as top, after mean-subtraction.

-Figure 1d-f, if this is data from 3 animals, do traces indicate the mean? Please show error bars/shadow. Are scale bars in f applicable to d and e?

These are not means but the power in the beta band from the L and R claustra, calculated over a scrolling window (see methods: window width: 10 seconds; scrolling step: 1 second). Fig 1d-f are excerpts from the 48hr recording shown in Figure 1c (at times indicated in that figure by black bars above the x-axis, labeled "d", "e", and "f"). Scale bar is the same for d, e and f, as indicated in legend, and correspond to the beta power after normalization as described in methods.

-The authors write "Sensory inputs such as light, sound or touch delivered to sleeping animals typically awaken them if sufficiently intense." I believe a reference should be added since authors will make a strong point later about applying stimulation that is not enough to awake the animals. A related point to this is that, the authors use light as an external stimuli, but do they think that the results will be preserved with other sensory modality? They should provide an experimental test that rule out this possibility, as it will make their claims stronger.

Our statement (sensory inputs such as ...) is a general description of the susceptibility of sleep to interruption by external stimuli, and of its greater threshold than during awake states. This description applies to Pogona, as observed by us (Shein-Idelson et al., 2016), and our light stimuli were calibrated empirically, to be in a range that did not awaken the animals.

We do not believe that our results concern light alone, but we did not explore other sensory modalities systematically; we chose light over sound or touch because light pulses were the simplest stimuli to deliver reliably across animals and experiments, uni- and bilaterally, and without repeated testing of threshold intensities. It will be interesting to test other modalities in future work.

-is the total amount of REM preserved across nights? Please quantify.

These measurements are now provided in Extended Data Fig. 2. Over all tested conditions, light pulses led to a small but barely significant ($p=0.032$) increase in total REM and a small but not significant ($p=0.32$) increase in the median duration of REM. Conditions of pulse delivery that led to a cessation of REM (e.g., short inter-pulse intervals) were followed by a temporary homeostatic increase in REM and decrease in SW, as detailed below.

-Figure 2d: please define what is considered “early” and “late” NREM/REM, is it first and second half of a given episode? Please describe in methods.

We have added the requested information in the methods section. For consistency with a previous publication and in response to reviewer 3, we renamed REM ‘REMp’ and NREM ‘SW’ throughout the manuscript. The added paragraph reads:

“Phase analyses and phase response curves”: In Fig. 2d,e,f and Fig. 5b,c,d,e, we define the phase (φ) of the pulse as ‘early SW (blue) if $\varphi \in [0, 0.25]$, ‘late SW’ (green) if $\varphi \in [0.25, 0.5]$, ‘early REM_P’ (orange) if $\varphi \in [0.5, 0.75]$, and ‘late REM_P’ (red) if $\varphi \in [0.75, 1]$, where $\varphi = 0$ is the onset of SW.”

-Figure 5a: methodology for eye occlusion is incomplete, is unclear how this was done and whether the animal displayed normal sleeping patterns after this manipulation. Please quantify and compare to regular sleeping nights (animals not blinded) the amount of REM/NREM overall, the duration of each cycle and over time, the number of transitions...

We now provide all these values in Extended Data Fig. 2, where we compare non-stimulated sleep, sleep under 1s pulses and sleep under 1s pulses with unilateral eye-cups. Light stimulation, with or without eye occlusion, led to a small increase of REM that nevertheless remained within normal sleep ranges. We observed no other significant effect on the animals’ sleep patterns.

Regarding methodology, we report in the methods section: “For monocular stimulation experiments, a black, 3-D printed plastic cup was secured to either left or right eye using silicon (Kwik-Sil™). Eye cups were attached ~30 min before starting an overnight recording, and removed the following morning.” Cups were attached and removed while the animals were awake, by hand and with no need for specialized tools.

For example, are firing rates of units changed during these conditions of eye closed? Since the authors have the data, please quantify.

-In general, firing rates results should be included for every experiment (i.e., light stimulation), to get a better understanding of how different parameters regulate individual cells.

In this study, our assessments of the sleep rhythm are population measurements on claustral field potentials, using beta power as a proxy for REM. Indeed, power in the beta band represents SN-production rates (see Fig. 1d in Fenk et al Nature 2023), where SNs each represent the simultaneous (within a few ms) discharge of single spikes from claustral units (Fig. 1e and Extended Data Fig. 1e, Fenk et al., 2023). The information about population firing rates is thus implicitly contained in our metrics.

-Figure 5g: this result is interesting but not convincing. Please provide further quantifications of this (not only the phase of the stimulus but the timing within the overall sleep for example) and separate or provide quantification for different animals, to rule out the possibility of inter-subject variability. Were all stimuli given for the same duration? Doing experiments where distinct durations of the stimulus are given, could provide insights into how the re-synchronization happen. One way would be to analyze units activity, as the authors state in the methods that they have clustered the data into single neurons using IronClust. If re-synchronization is happening due to an internal mechanism, as the authors suggest, it should be quantifiable with cross-correlograms of neurons from each hemisphere.

We provide below (a-c) a breakdown of Figure 5g by animal (b) and by time within sleep phase (c). We observed long delays to re-synchronization in all tested animals within the critical-phase range ($\varphi \sim 0.25$) (b). We did not observe any effect of time-of-the-night on delay-to-resynchronization (c).

Response to Reviewers Figure 2. Breakdown of time at which the two sides re-sync by animal and time of the night.

a. Same as Fig. 5g. *b.* Same as *a* with experimental animal indicated in color. Data pooled from 162 trials across 13 recordings and 4 animals. *c.* Same data as in *a* but plotted as function of the time at which the unilateral pulse was delivered during the night.

All stimuli in Figure 5 were 1-second light pulses. We clarify these points in the legend of Figure 5g that now reads:

“Data pooled from 162 1s-long light pulses across 13 nights in 4 animals”.

Concerning the suggested experiments with distinct durations of monocular pulses: although possible, we think that this analysis would not be informative about the mechanisms of resynchronization, for the following reason. The reset and resynchronization must happen well upstream of the claustrum: activity in the claustrum is driven by that in the isthmus (see Fig 4, Fenk et al, 2023), and activity in the isthmus is presumably itself downstream of the putative CPGs (this paper), which we believe to be

located in the brainstem. We therefore believe that cross-correlograms of L and R claustral units, although possible, would not provide easily interpretable information about interactions between L and R CPGs, located many synapses upstream.

-In the methods, the authors say that “the lizards were placed into the arena and left to sleep and behave naturally overnight. They were returned to their home terraria the next day, when they received food and water. Experiments were performed at room temperature of around 21.5 °C.” Does this mean that during the time that data is being analyzed (sleeping in the arena) animals were water and food deprived? Please clarify.

Although we carried out some continuous 24 or 48-hr recordings, our experiments typically started in the late afternoon, 1-2 hrs before the automated room lights (on a 12hrs ON - 12hrs OFF cycle) went off, and ended a few hours after lights came on again in the morning. Shortly after the time when recordings started, the animals typically prepared for sleep: they took on a characteristic posture, rested their head on a foreleg or the substrate, closed their eyes, and progressively fell asleep. Pogona sleep is sustained, typically lasting 8 to 11 hrs, and uninterrupted under our experimental conditions (and 12L-12D circadian entrainment). During that time, the animals showed no signs of awakening: while they occasionally repositioned a limb or their head, the animals did not engage in any locomotion, food-seeking, feeding or any other active behavior. Pogona is a diurnal reptile, and naturally undergoes extended periods of food and water restriction. Thus, we supplied no food or water during the night recordings.

Referee #2 (Remarks to the Author):

This manuscript explores the hypothesis that *Pogona*'s NREM-REM ultradian rhythm is produced by a central pattern generator (CPG). Specifically, the authors test whether the ultradian rhythm can be reset in a phase-dependent manner and whether it exhibits entrainment by light pulses. They also used a short light pulse delivered to a single eye to establish that sleep is generated by paired rhythm-generating circuits linked by functional excitation. This is innovative work in an understudied animal model, and there are interesting features of *Pogona*'s sleep [e.g., a short (~2 min) period and high

regularity] that are of general interest to sleep researchers and that motivate the use of this animal model to address these questions. The demonstration that “REM can be shortened but typically not extended beyond a natural limit (p.5)” is elegant and compelling. However, I have some reservations about some of the interpretation/conclusions, and I feel that some aspects of the manuscript require more context. Below, I have detailed my questions and concerns regarding the manuscript.

Major concerns:

1. The second paragraph of the introduction describes two hypothesized mechanisms for REM sleep generation, and the third paragraph states that both of these mechanisms require “reciprocally inhibitory circuits.” However, the term “reciprocal interaction” described in the first mechanism refers to inhibitory- excitatory coupling between monoaminergic and cholinergic neurons, respectively, in the conceptual and mathematical model proposed by McCarley and Hobson. These two hypothesized mechanisms actually generate cycles using distinct dynamic structures (hysteresis loop vs. limit cycle) as described in ref 27 (Diniz Behn et al., 2013). The description of the reciprocal interaction circuit should be corrected to clarify the difference between the mechanisms.

We thank the reviewer for this comment. We modified the text to use terminology consistent with Diniz Behn et al., 2013, using “mutually inhibitory” to refer to inhibitory-inhibitory couplings and “reciprocal interactions” for excitatory-inhibitory couplings. We also clarified the beginning of the third paragraph to make clear the existence of the two mechanisms. All changes are highlighted in the manuscript and summarized below:

In the Abstract:

“The mechanisms underlying the mammalian ultradian sleep rhythm—the alternation of rapid-eye-movement (REM) and slow-wave (SW, also non-REM) (NREM) sleep states—

are not well understood but likely depend, at least in part, on **reciprocal** interactions between groups of neurons in the brainstem”

In the Introduction:

“Although the mechanisms underlying mammalian NREM-REM alternation are unknown (Weber, 2017), two classes of hypotheses exist, that rest on the identification of brainstem neurons with antiphasic activity during sleep, and on models inspired by these results. In the first class of hypotheses (Hobson et al., 1975), monoaminergic neurons in the locus coeruleus (LC) and dorsal Raphe inhibit cholinergic neurons in the pontine tegmentum with excitatory back projections. In corresponding models, **these reciprocal interactions produce an** alternating and sustained rhythm during sleep due to self-inhibition of the monoaminergic neurons releasing the cholinergic neurons to trigger a REM episode. Although these models have limit-cycle solutions qualitatively consistent with experiments (McCarley & Hobson, 1975), they rely on neural connections that have not been confirmed experimentally (Jones et al., 1977; Lu et al., 2006; Sastre et al., 1981; Shouse & Siegel, 1992; Weber, 2017). This led to a second class of models that rely on two key features **to produce a hysteresis loop (Diniz Behn et al., 2013 ~~Boeth & Diniz Behn, 2014~~)**: the existence of **reciprocal or** mutual inhibitory circuits, suggested by anatomical data in the brainstem and elsewhere (Boissard et al., 2002; Clement et al., 2011; Diniz Behn et al., 2013; Lu et al., 2006) to stabilize each sleep state; and the existence of a separate, slow-evolving and so-far-unknown mechanism, termed “REM pressure”, that accrues during wake, SW or both, and triggers the transition between states (Benington & Heller, 1994; Weber, 2017).

Although different in **their dynamic structures, detail** these competing models agree on **inhibitory feedback being a key circuit element ~~the need for reciprocally inhibitory circuits~~** to generate sleep-state alternation. Circuits **that generate** alternating outputs **without requiring alternating inputs**, often called central pattern generators or CPGs, are well known in motor systems”

In the Discussion:

*“Whereas many investigations of the ultradian sleep rhythm proposed the existence of **mutually inhibitory synaptic projections reciprocal interactions** (e.g., “flip-flop”, (Dunmyre et al., 2014; Lu et al., 2006)), they did not explicitly link them to CPGs.”*

2. Ref 7 (Shein Idelson et al. , 2016) indicates that the period of the ultradian rhythm in Pagona varies over the course of the night as it does in humans. However, the trial-to-trial variability (e.g., in Fig 1g) seems very small. Were additional (time of night?) criteria used to select the cycles included in these analyses?

We added the following precisions to the Methods section “Light-pulse experiments”:

“In all non-entrainment, single-pulse experiments during sleep (Fig. 1, Fig. 2, Fig. 5, Extended Data Fig. 4), light pulses were delivered between hours 2 and 10 of the recording (i.e., between approx 8pm and 4am). We manually inspected and discarded the few instances in which the sleep rhythm preceding the light pulse was particularly irregular, because it affected our capacity to estimate the phase of the pulse. Hence, 9.15% (or 66 out of 721 in all) of the pulses (all durations) were excluded. No additional criteria (time of night, phase of the pulse, or length of the cycle) was used for this selection.”

3. Ref 7 also mentions that the period of the ultradian rhythm is temperature dependent. The issue of temperature dependence in these rhythms should be addressed, at least in the Discussion.

We added a note, in the context of our discussion of and comparison with the mammalian infraslow rhythm (Lüthi and colleagues). This now reads:

“Note that the period of the noradrenergic infraslow rhythm in mice is shorter (30 – 50s) than that of Pogona’s SW-REM rhythm (around 130s at ~ 21.5 °C), but the difference could, in principle, be accounted for by the differences in body temperature and the known temperature dependence of the sleep rhythm frequency in Pogona (Shein-Idelson et al, 2016).”

4. In mammalian species, there is evidence for a homeostatic pressure for REM sleep (e.g., Endo et al., AJP, 1998). Is there similar evidence for REM homeostasis in Pogona? Specifically, does the alternation between high and low claustral beta show any evidence of modulation by REM homeostasis or change in response to a homeostatic challenge?

The question of REM homeostasis could in fact be explored specifically and precisely, thanks to our sleep-rhythm reset findings. By stimulating sleeping animals with repeated light pulses at sufficiently short intervals, REM could be suppressed for long epochs of many minutes. Upon cessation of the stimuli, REM returned and occupied an increased fraction of the subsequent sleep cycles, consistent with homeostatic regulation of REM. This is now detailed in Extended Data Fig. 4 , as shown below:

(New) Extended Data Fig. 4. REM_p sleep homeostasis.

a, Train of 1s-long light pulses delivered at a short inter-pulse interval (IPI) of 80s. Such a short IPI fails to entrain the ultradian rhythm, and suppresses REM_p for several minutes (top). Upon cessation of the stimuli, REM_p resumes and occupies a larger fraction of the sleep cycle than before stimulation (bottom left, $**P = 0.0078$, $W = 0$), with longer REM_p average duration (bottom middle, $*P = 0.0234$, $W = 2$) and shorter SW average duration (bottom right, $*P = 0.0156$, $W = 1$); $n = 8$ experiments; 25 min preceding (pre) and following (post) pulse trains were used for statistical comparison. Wilcoxon signed-rank paired tests. $**$: $P < 0.05$; $*$: $P < 0.1$. These results are consistent with a recent study reporting sleep homeostasis in *Pogona*⁵⁷. **b**, Same as in **a** but with light pulses applied at IPI = 180s, causing reliable entrainment (see Fig. 3). This regime is accompanied by no alteration of the percentage of post-stimulation REM_p (bottom left, $P = 0.375$, $W = 18$), average REM_p duration (bottom middle, $P = 0.1602$, $W = 13$) or average SW duration (bottom right, $P = 0.4316$, $W = 19$); $n = 10$ experiments. **c**, Two consecutive trains of 60s and 80s-long light pulses, separated by 60 min, and each consisting of 10 pulses delivered every 60s and 80s, respectively. Both trains are included in the quantifications of **a** and indicated by red lines. Suppression of REM_p during the light pulses is followed by a rebound, visible as an increase in REM_p and simultaneous decrease in SW duration, slowly returning to baseline levels after stimulation (bottom panel). Open circles indicate long (>240s) SWS or REM periods. **d**, Quantification of the data in **c**, comparing the hour

preceding the first pulse train (I) with the hour following it (II), and the three hours following the second train (III-V). Left panel (from left to right): ** $P = 0.0060$, $U = 196$; * $P = 0.0225$, $U = 212.5$; $P = 0.0555$, $U = 232.5$; $P = 0.9361$, $U = 356$. Right panel (from left to right): $P = 0.0891$, $U = 447$; * $P = 0.0185$, $U = 466.5$; $P = 0.5270$, $U = 290$; ** $P = 0.0066$, $U = 198$. Mann–Whitney U-tests. ** and * as in **a**.

We also updated Fig. 3f to include all 80s IPI trials, rather than only those in which at least 4 SW/REM cycles occurred (previous version). The figure thus now incorporates trials resulting in extended epochs of sustained SW and suppressed REM. We refer to the new EDF 4 in the context of the results shown in Fig 3:

*“These effects of entrainment on period were caused mainly by an elongation of SW (Fig. 3f); REM_p duration remained bounded to its maximum natural duration over the range of entraining IPIs (Fig. 3e). **When the IPI was very short (80s), the extension of SW (Fig. 3f) was accompanied by a concomitant reduction or even suppression of REM_p. This suppression was typically followed by a rebound of REM_p and a reduction of SW after the stimuli ceased (Extended Data Fig. 4).** The greater elasticity of SW compared to that of REM_p is consistent with the results in Fig. 2 (note the asymmetry around $\varphi = 0.25$ of the phase response curve; inset in Fig. 2e).”*

5. Relatedly, a correlation between the duration of a REM episode and the duration of the following NREM episode has been reported in mammals including rats, cats, monkeys, and humans (e.g., Vivaldi et al., J Neurophysiol, 1994; Benington and Heller, AJP, 2000; Barbato et al., SLEEP, 1998). Does this association hold in the ultradian cycles observed in Pagona?

We did look for such potential correlations between the durations of adjacent REM and SW epochs, testing all possible relationships. We summarize these results in the figure below. We analyzed consecutive REM and SW epochs that lasted between 30 s and 120 s. We observed, on average, a negative correlation between a REM episode and its preceding and following SW episodes, and a positive correlation with the next REM episode (a). This trend is preserved across 34 normal (i.e., with no light stimulation) sleep recordings, and involves not only the following SW and REM episodes, but extends over multiple cycles (b). These negative correlations between REM and surrounding SW partly explain the very stable sleep rhythm of *Pogona*.

Response to Reviewers Figure 3. Correlation between a REM episode and its surrounding SW and REM episodes.

a, Duration of REM episodes from a single recording (x-axis) plotted against preceding SW (left, Pearson's $r = -0.290$, $***P = 0.00000067$, $n = 283$), following SW (middle,

*Pearson's $r = -0.411$, $***P = 0.00000000000016$, $n = 272$), and following REM (right, Pearson's $r = 0.186$, $***P = 0.0018$, $n = 281$) episodes. Red lines are linear regressions. **b**, Correlations calculated as in **a**, pooled across 34 recordings from 20 animals, comparing the duration of REM episodes with surrounding REM and SW episodes. Red indicates averages. Positive correlations with subsequent REM episodes and negative correlations with subsequent SW episodes persist for multiple cycles.*

6. The authors interpret the middle of NREM as a “critical phase of the sleep cycle” that separates phase advancing/delaying effects of light pulse. Can this interpretation be distinguished from the existence of a REM refractory period (cf. LeBon, Sleep Medicine, 2020) during which REM-like behavior cannot be initiated? The presence of a refractory period would be supported by the results in Fig 2f showing the response to a 90s light pulse when, in the top panel, the beta activity is eventually triggered.

We thank the reviewer for this comment. The description of apparent REM refractoriness might indeed apply to early SW, when light pulses typically fail to trigger a new REM (and cause a phase delay). The critical phase we describe would then fall at the end of this refractory period, after which a light pulse can trigger REM again (and cause a phase advance). As pointed out by the reviewer, the result in Fig. 2f (top panel) is consistent with such an interpretation, suggesting a refractory period in the order of 30s. Note, however, that the inter-quantile range is wide, and that if pulses typically do not trigger REM in this interval, a few occasionally do. In our view, the existence of a REM refractory period is thus compatible with, and could provide a potential mechanistic account for our phase-based description, but this refractoriness is relative rather than absolute. We added a mention of this in the discussion that reads:

*“... Finally, the phase dependent action of a light pulse identifies the middle of SW as a critical time when the effect of an external input switches from phase delay to advance. **This result is compatible with considering the first half of SW as a (relative) REM_P-refractory period...**”*

7. There has been a recent focus on the role of an infraslow rhythm modulating the microstructure of sleep in rodents (e.g., Stucynski et al., Curr Biol, 2022; Kjaerby et al., Nat Neurosci, 2023). Given the timescale of the ultradian oscillation in *Pogona*, it would be interesting to know if there is any evidence for an infraslow rhythm modulating sleep in this amniote.

We thank the reviewer for this comment, which converges with one made by reviewer #3 and is indeed relevant. We do not at present know whether a similar rhythm exists (in LC for example) but this can be tested. We now address the potential relationship in the discussion, in an additional paragraph that reads:

“Alternatively, the reptilian rhythm might correspond to a recently identified mammalian noradrenergic infraslow rhythm in the locus coeruleus (LC) that partitions SW sleep into two states of low and high arousability (Lecci et al., 2017; Osorio-Ferero et al., 2021; 2024; Kjaerby et al., 2022), defining periodic epochs of potential entry into REM sleep. By this hypothesis, the reptilian rhythm would have been co-opted in mammalian evolution and become a periodic gate that defines when transitions into REM can, but do not necessarily happen. This in turn would have enabled the duration of each mammalian sleep phase to be under additional levels of control (e.g., REM “pressure”). Note that the period of the noradrenergic infraslow rhythm in mice is shorter (30 – 50s) than that of *Pogona*’s SW-REM rhythm (around 130s at ~ 21.5 °C), but the difference could, in principle, be accounted for by the differences in body temperature and the known temperature dependence of the sleep rhythm frequency in *Pogona* (Shein-Idelson et al, 2016).”

Lecci, S., Fernandez, L.M.J., Weber, F.D., Cardis, R., Chatton, J.-Y., Born, J., and Lüthi, A. (2017). Coordinated infraslow neural and cardiac oscillations mark fragility and offline periods in mammalian sleep. Sci. Adv. 3, e1602026.

Osorio-Forero A, Cardis R, Vantomme G, Guillaume-Gentil A, Katsioudi G, Devenoges C, Fernandez L.M.J., Lüthi A., 2021, Noradrenergic circuit control of non-REM sleep substates.

Curr Biol, 31, 5009-5023.e7, ISSN 0960-9822, <https://doi.org/10.1016/j.cub.2021.09.041>.

Kjaerby C, Andersen M, Hauglund N, et al. Memory-enhancing properties of sleep depend on the oscillatory amplitude of norepinephrine. Nature Neuroscience. 2022, 25(8):1059-1070. DOI: 10.1038/s41593-022-01102-9. PMID: 35798980; PMCID: PMC9817483.

A. Osorio-Forero, G. Foustoukos, et al. Noradrenergic locus coeruleus activity functionally partitions NREM sleep to gatekeep the NREM-REM sleep cycle. Preprint at bioRxiv <https://doi.org/10.1101/2023.05.20.541586> (2024)

Minor concerns:

1. The Introduction should include more background on Pogona's sleep behavior for readers not familiar with previous work (e.g., ref 7). It would be helpful to review the diurnal behavior pattern and occurrence of a consolidated sleep period.

We added the following at the beginning of the results section:

“Sleep in Pogona occurs at night and consists of equal-length epochs of SW and REM-like activity (REM_P) alternating every minute at room temperature for 8-10 hours (Shein-Idelson et al., 2016).”

2. The term “reciprocal interaction” is often used to describe the excitatory-inhibitory coupling in the McCarley-Hobson model. It would be best to avoid this terminology when referring to mutually inhibitory coupling to avoid confusion.

We thank the reviewer for this comment. We have changed the text to use terminology consistently as described in our response above (to Major concern 1).

3. The interpretation of high beta power as a marker of REM sleep is not intuitive to researchers familiar with mammalian sleep. Some of the justification for this marker provided in refs 7 and 9 should be included here.

Pogona sleep is characterized by the regular alternation of two main electrophysiological states that can be distinguished using the spectral features of the LFP, as recorded from the dorsal ventricular ridge (DVR) and claustrum: epochs of low-frequency activity (< 4Hz, or high delta power), reflecting the irregular occurrence of SWRs generated in the claustrum during SW, and periods with faster awake-like activity that are captured well by increased power in the 10-30 Hz (beta) range, associated with rapid eye movements (REM). These features have been described in detail in Shein-Idelson et al., 2016; Norimoto, Fenk et al., 2020, and Fenk et al., 2023.

We have previously shown that REM activity in the claustrum consists of brief synchronized firing bursts, coincident with sharp negative deflections (SNs) in the LFP that occur at about 20 times/s, matching the observation that REM in the claustrum is dominated by LFP power in the 20 Hz (beta) range (Fenk et al., 2023). We include below Extended Data Fig. 2a of that paper, showing claustral LFP traces with their band spectrograms and illustrating the usefulness of beta as a marker for REM. Note,

however, that beta may not be the most appropriate choice as a marker for REM in other (e.g., cortical) regions that dominate in (surface) EEG recordings frequently used for sleep-stage scoring in mammals.

Response to Reviewer Figure 4. Legend of Extended Data Fig. 2a in

Fenk et al., 2023:

α , LFP traces from the paired recording shown in Fig. 2a, with their band spectrograms (0.1–100 Hz). Note beta band used to define onset and offset of REM.

4. In the transitions between periods of high and low beta associated with REM and NREM sleep, respectively, what is the threshold used distinguish between states, and how is it selected?

We describe the detection of REM and SW (NREM) from beta power in detail in our Methods. We have renamed the subsection to “Beta power and REM_P / SW detection” for clarity.

In short: we defined thresholds automatically using a Gaussian mixture model fitted to the log beta distribution and merged together epochs defined by successive threshold crossings separated by intervals < 15s. For awake experiments, typically shorter because animals tolerated only one round of light-pulse entrainment per session, thresholds were defined by hand, such that they ensured the accurate identification of SN-rich epochs, characteristic of awake activity (and REM) in the claustrum. We used the same method in Fenk et al., 2023.

5. It sounds like the first sentence on p. 4 (“Because the animals...lights go out.”) is intended to describe anticipatory behavior resulting from entrainment. However, this meaning should be clarified since this interpretation of eyes closing spontaneously is not obvious. This idea was stated more clearly in ref 7.

We did not fully understand this remark. Eye-closure is indeed, together with settling in one position and a progressive decrease of postural tone, an external mark of the animals’ progressive entry into sleep in anticipation of the turning off of the room lights (Pogona is diurnal), after circadian entrainment. (Pinealectomized Pogona, for example, does not show any locking of sleep/wake to external time — pers. communication.)

Hoping that this may clarify our statement, we replaced “their eyes typically start closing spontaneously” with:

“Shortly before the lights go out in the evening, the animals (because they are entrained by a 12h light - 12h dark rhythm) typically settle in one place, display decreasing postural tone and spontaneously start closing their eyes”

6. P. 4 The authors describe the oscillator circuit as “activated and inactivated by circadian clock.” This implies both a positive and negative action by the clock at different times/phases. Is this the intention, or could this be regulation be better described as “gating by” the circadian clock? If the former, is there evidence for positive and negative action?

Thank you. The reviewer is correct: gating better describes our thoughts. We replaced the original statement with:

“we hypothesized that this rhythm is due to the action of an oscillator circuit, gated by the circadian clock”.

7. Fig 3b: how is preferred phase defined?

This described the phase of the cycle at which a pulse fell when locking was tight (narrow unimodal distribution). We have updated the figure legend to read:

“b. Distributions of pulse phase during entrainment. Scale bar indicates a density of 1. Number of pulses per IPI between 81 and 131. Narrow distributions indicate phase-locking and good entrainment (IPIs between 120 and 200s).”

8. The Methods should specify that the lizards were entrained to a 12:12 Light:Dark cycle with time of lights on and intensity of light during the light period.

We now include a sentence in the Methods section that reads:

“Lizards were entrained to a 12:12 Light:Dark cycle ($t_{off} = 18:00/19:00$, winter/summer and $t_{on} = 06:00/07:00$, winter/summer). Light intensity during the artificial day, measured at the eye, was 15.5 lx (same intensity as for the light pulses)”

9. There was minimal time for habituation to the eye cups used for the monocular stimulation experiments. Was there any evidence of stress in the animal and/or abnormal sleep behavior associated with the use of eye cups?

We thank the reviewer for this important question. The animals were indeed (and perhaps surprisingly) very tolerant of the eye cups. Signs of possible discomfort, if present, were typically restricted to the first minutes following attachment of the cups. The animals would soon after start to engage in their normal behavioral routines, and we did not observe any abnormal sleep behavior.

10. In the REM detection experiments, the authors note that “the animals generally tolerated only one round of light pulse entrainment per session.” What does “tolerated” mean in this context? Could the REM-like and NREM-like behavior only be elicited once? What might this mean for the circuit?

This “tolerance” refers to awake experiments only (Fig. 4), in which we exposed the animals to a regime of alternating light and dark pulses to try and entrain their brain activity (movie 2). A successful awake entrainment experiment thus required the (awake) animal to remain relatively still for long periods ($\geq 10 \times 120s$, i.e., more than 20 mins at least for a single run) and not be distracted by events in its surroundings. This state could not be maintained for very long, restricting the possible duration of awake entrainment experiments to typically 30-50min. After this time, the animal would generally wander off and not “tolerate” other rounds of light/dark stimuli. Also, if the entrainment period of the light/dark alternation regime deviated much from that of their natural ultradian sleep (either being too short, or too long), the animals (always allowed to behave freely throughout) were more likely to become agitated, forcing us to abort an experimental session.

These observations may suggest the existence of a state that allows the transient activation of a dormant ultradian sleep oscillator, and yet is sensitive to inputs from circuits associated with heightened arousal and overt behavior. This potential higher-order interaction between brain states is very interesting, but probably beyond the scope of the present study.

11. Phase analyses and phase response curves: Is there a typo in “band pass filtered the beta time series by 0.00277 – 0.16666Hz (corresponding to 1/6 minutes and 1/1 minute)”? Not sure what this is saying.

Thank you for the comment. We rephrased and hopefully clarified, to read:

“We first filtered the beta time series in the band 0.00277-0.16666Hz using a Butterworth filter. This band preserves the range of timescales relevant to the beta cycle (0.00277 Hz = 6 minutes⁻¹, 0.16666 Hz = 1 minutes⁻¹). We then extracted the Hilbert transform, using the `scipy.signal.hilbert` function in Python.”

12. Top of p. 17: Choice to evenly divide interval between NREM and REM seems odd in situations where NREM and REM are not evenly divided across the ultradian cycle. Authors should comment on the appropriateness of this choice.

Normalizing phase to the range $\varphi \in [0,1]$ is common in studies of neurobiological oscillators (e.g. Canavier 2015, Curr. Opin. Neurobiol.). For simplicity, we phase-shifted this range so that $\varphi = 0$ corresponds to the transition from high to low beta power, that is, to the beginning of SW sleep.

While sleep in Pogona is typically evenly divided between SW and REM (see new Extended Data Fig. 2, Duty cycle), our method does not require it. Even in situations where SW and REM are not equal in duration, the mathematical properties of the Hilbert transform combined with the band-pass filtering of beta result in identifying the sharp transition from low to high beta (SW to REM sleep) as $\varphi = 0.5$. This property allows us to detect the phase advances and delays (as in Fig. 2d,e) even when the duty cycle deviates from 50%, due to the light stimuli. We illustrate this property with synthetic data in the figure below.

Response to Reviewers Figure 5. Phase estimation under uneven SW and REM durations.

Top: synthetic beta data showing 5 cycles where SW and REM match in duration, 5 cycles where SW is three times longer, and 5 cycles where REM is longer. Bottom: phase estimation (black line). The transition of REM-to-SW are consistently mapped to $\varphi = 0$ (cyan dots) and the transition of SW-to-REM to $\varphi = 0.5$ (pink dots) despite the altered duty cycle.

We have clarified this point in our Methods section “Phase analyses and phase response curves”:

“The resulting time series maps the beginning and end of SW sleep to 0 and 0.5 and the beginning and end of REM_P to 0.5 and 1. Due to the properties of the Hilbert transform applied to the smoothed (band-passed) signal, this mapping of 0.5 to the transition of low to high beta remains true even when the duty cycle is not 50%.”

Referee #3 (Remarks to the Author):

The paper presents convincing evidence that the biphasic sleep pattern observed in the Pogona lizard is governed by a CPG-type oscillator process. The experiments and reported results are aligned with the typical manipulations conducted to determine the existence of a CPG such as phase shifting and entrainment. This is a very interesting finding pointing to novel hypotheses about the neural mechanisms controlling sleep and transitions between sleep states. The results provide new insights into processes that may govern and contribute to sleep patterning in other animals, including mammals. The sleep community will be particularly interested in these results but they should also have broader appeal as an interesting example of an evolutionarily simpler process to govern sleep.

I am skeptical that REM and NREM state alternation in mammals is governed by the same mechanism. Particularly given the direct effect of light on the identified CPG process shown here while in mammals circadian modulation from the SCN to sleep-promoting centers is polysynaptic and not entirely identified.

It is important to distinguish the circadian- and ultradian-rhythm reset and entrainment mechanisms, which we do not believe to be the same. The mechanisms of action of light onto the ultradian CPG are not known, but likely independent of the SCN and the melanopsin-RGC pathways already known to underlie circadian entrainment in vertebrates. Rather, we believe that light input (i.e., the resetting inputs evoked by light pulses as in our experiments) to the ultradian CPG are mediated by the retina via the optic tectum (reptilian homolog of the SC) and/or retino-recipient thalamic nuclei. Thus, while it is not clear yet whether the mechanisms underlying the ultradian REM/SW alternation are similar in reptiles and mammals, we do not believe that our data are inconsistent with the existence of common mechanisms (at least in part) across amniotes.

However, the reported CPG-process could be related to mechanisms responsible for recently identified slow (~50s) rhythmic activity in hypothalamic and brainstem areas that appears during NREM sleep. Specifically, recent studies have shown that the EEG during NREM sleep in both humans and mice shows a pronounced infraslow modulation in the sigma (10-15 Hz) power range (see PMC5298853, PMC8752505, PMID: 34648731 for example). Transitions out of NREM sleep to both wake and REM sleep are synchronized with this infraslow sigma power rhythm. This alternation in brain states marked by activity in different frequency bands during sleep on a slow time scale is not entirely dissimilar from the Pogona sleep patterning. The mechanisms responsible for the infraslow sigma band rhythm in mammals is not known but may also be neural in origin, as hypothesized for the mechanism of the CPG process reported here. I think it would increase the impact of the results for the authors to include a discussion of a potential relation to this sigma band rhythm; the Pogona beta power sleep rhythms may perhaps be an evolutionarily earlier mechanism for rhythmic patterning of different brain activity levels during sleep that shape the well-known NREM-REM alternation.

We thank the reviewer (and reviewer 2) for this comment with which we entirely agree. An interesting possibility, indeed, could be that the mammalian infraslow/locus coeruleus rhythm represents an ancestral biphasic sleep rhythm as that identified in Pogona (likely to better represent the common amniote ancestor). (Note however that we do not detect arousals in Pogona at the SW-REM transition.) Sleep in mammals might have evolved in such a way that the transition to REM (or wake) remained locked to this rapid oscillator, while the duration of REM (and of SW) became dependent on new factors, independent of the oscillator. In this interpretation, the ancestral rhythm or CPG became a permissive gate for sleep-state transition, but without the transition obligation observed in Pogona. This hypothesis leads to testable experiments in Pogona, such as whether LC's noradrenergic output is periodic and matches the ultradian CPG rhythm. We will test this in future experiments.

We added a paragraph in the discussion that reads:

“Alternatively, the reptilian rhythm might correspond to a recently identified mammalian noradrenergic infraslow rhythm in the locus coeruleus (LC) that partitions SW sleep into two states of low and high arousability (Lecci et al., 2017; Osorio-Ferero et al., 2021; 2024; Kjaerby et al., 2022), defining periodic epochs of potential entry into REM sleep. By this hypothesis, the reptilian rhythm would have been co-opted in mammalian evolution and become a periodic gate that defines when transitions into REM can, but do not necessarily happen. This in turn would have enabled the duration of each mammalian sleep phase to be under additional levels of control (e.g., REM “pressure”). Note that the period of the noradrenergic infraslow rhythm in mice is shorter (30 – 50s) than that of Pogona’s SW-REM rhythm (around 130s at ~ 21.5 °C), but the difference could, in principle, be accounted for by the differences in body temperature and the known temperature dependence of the sleep rhythm frequency in Pogona (Shein-Idelson et al).”

Lecci, S., Fernandez, L.M.J., Weber, F.D., Cardis, R., Chatton, J.-Y., Born, J., and Lüthi, A. (2017). Coordinated infraslow neural and cardiac oscillations mark fragility and offline periods in mammalian sleep. Sci. Adv. 3, e1602026.

Osorio-Forero A, Cardis R, Vantomme G, Guillaume-Gentil A, Katsioudi G, Devenoges C, Fernandez L.M.J., Lüthi A., 2021, Noradrenergic circuit control of non-REM sleep substates.

Curr Biol, 31, 5009-5023.e7, ISSN 0960-9822, <https://doi.org/10.1016/j.cub.2021.09.041>.

Kjaerby C, Andersen M, Hauglund N, et al. Memory-enhancing properties of sleep depend on the oscillatory amplitude of norepinephrine. Nature Neuroscience. 2022, 25(8):1059-1070. DOI: 10.1038/s41593-022-01102-9. PMID: 35798980; PMCID: PMC9817483.

A. Osorio-Forero, G. Foustoukos, et al. Noradrenergic locus coeruleus activity functionally partitions NREM sleep to gatekeep the NREM-REM sleep cycle. Preprint at bioRxiv <https://doi.org/10.1101/2023.05.20.541586> (2024)

Major point:

In my opinion, the attempt to relate the alternation of the high and low beta power sleep states in the *Pogona* to NREM-REM alternation in mammals is made a little too strongly. Until further experiments are done that replicate key properties of NREM-REM alternation in mammals, such as response to sleep deprivation, the competition of NREM and REM sleep rebound after deprivation, homeostatic regulation, etc, it's better if the authors don't conflate the two. For example, the 2nd paragraph in the Introduction reviewing hypotheses for mammalian NREM-REM alternation may be better placed in the Discussion so that the relationship between the presented phenomena and mammalian NREM-REM alternation is not immediately presumed. Also,

on pg 10, end of middle paragraph, the existence of REM-on and REM-off neurons in mammals should not be related to the putative CPG in the Pogona.

We thank the reviewer for these comments, and hope that some of the additional evidence presented below addresses them.

Homeostatic regulation of sleep and REM:

We have added evidence for homeostatic competition between REM and SW (see Extended Data Fig. 4, also shown in response to Reviewer 2's major point 4 above), exploiting the fact that certain light stimulation regimes presented during sleep (e.g., light pulses presented at rates too fast to enable reliable entrainment) reduce or suppress REM. Upon recovery from such imposed epochs of REM suppression (e.g., 5-10 sleep cycles), REM duration was increased over several tens of minutes, at the expense of SW (see Extended Data Fig. 4, and response to major comment 4 of Reviewer 2 above).

In addition, an independent study explored more general aspects of sleep deprivation, providing evidence for sleep homeostasis in Pogona (Hatori, Yamaguchi et al., 2024).

Thus, these and other key properties of mammalian sleep (see Shein-Idelson et al., 2016) seem to apply to Pogona as well.

This does not prove that REM/SW in mammals and reptiles are homologous (as in deriving from common ancestry), but we also do not know what such proof might look like. Our hope is that, as our mechanistic understanding of the production of sleep in several key amniotes increases, the various degrees of equivalence and dissimilarity can be better evaluated and the most parsimonious explanations for the evolution of sleep in tetrapods and amniotes determined.

Changes to the introduction:

*We would like to retain the presentation of possible models for REM/SW alternation in mammals (par 2 of introduction) because this classic work sets the scene for thinking about CPGs in the context of sleep, which we think is directly relevant. We are careful to state, in the following paragraph: “We thus hypothesized that sleep’s ultradian rhythm may be, **at least in Pogona**, the by-product of a CPG circuit, possibly evolutionarily related to brainstem circuits for motor control”.*

Changes to the discussion:

In response to the reviewer’s request, we have taken out the mention of REM on and REM off neurons in the mammalian brain (last sentence of par 2, P10). The sentence now reads:

“This view is consistent with the tight regulation we observed for REM_p duration.”

Sena Hatori, Sho T. Yamaguchi, et al. Sleep homeostasis in lizards and the role of cortex. Preprint at bioRxiv doi: <https://doi.org/10.1101/2024.07.31.605950> (2024).

In their previous paper Fenk et al Nature 2023, the authors use the notation REM_p for the sleep state with high beta activity and explicitly state that the NREM and REM “nomenclature is descriptive, and does not necessarily imply, for lack of knowledge at this point, functional or mechanistic identity with mammalian sleep states.” To better avoid the conflation with mammalian NREM-REM alternation, they should likewise use that nomenclature here and make the same statement.

We have now introduced the same nomenclature (SW and REM_p throughout) as in our previous paper, and the same explanatory statement in the methods section. It reads:

“Note that the nomenclature (REM_P and SW) is descriptive, and does not necessarily imply, for lack of knowledge at this point, functional or mechanistic identity with mammalian sleep states.”

Comment on nomenclature:

We note that sleep is highly polymorphic even among mammals, and that significant differences exist between sleep across mammalian species that probably have both different functional consequences and different mechanistic explanations (e.g., consolidated sleep phase in humans, with its sustained and relatively regular REM/SW cycling, vs., fragmented / irregular sleep in certain rodents; absence of rapid-eye-movements in moles and other mammals; or the association of penile erections and REM sleep in most male mammals studied, but its association with SW in armadillos; this list is not exhaustive).

We note also that there exists no objective evidence (mechanistic or otherwise) for the equivalence of sleep states between mammals and birds (the latter being endothermic reptiles) although this equivalence (by convergence) is typically assumed in the literature. Consequently, some sleep researchers (cf Rattenborg & Ungurean, Trends in Ecol. and Evol., 2022) suggested that we should specify the species to which the terms REM and NREM are applied (e.g. REM_{Ho-sa}, REM_{Ra-no} or REM_{Co-li} for REM in Homo sapiens, Rattus norvegicus and Columba livia, respectively). This terminology, however, has not been adopted by the field, which is why its exclusive use for Pogona appears to us as singling out this model system. We will nevertheless comply, as we did before, and thus name REM in Pogona REM_P.

Rattenborg, N. C., & Ungurean, G. (2022). The evolution and diversification of sleep. Trends Ecol Evol. <https://doi.org/10.1016/j.tree.2022.10.004>

The CPG-type oscillator process responds directly to light and all manipulations of the CPG process are in relation to the action of external light on that process. Without knowing what the CPG process is, it may well be possible that there are other stimuli that can perturb the process and have different phase shifting or resetting effects. The authors should make it clear in statements that interpret the function of the CPG process that the results show only the effects of light pulses on the CPG process. For example, on pg 5, last sentences of the 1st paragraph, the authors state “In other words, REM can be shortened but typically not extended beyond a natural limit, whereas NREM can be both shortened and lengthened.” Here, this is shown only for the effect of a light pulse.

This is correct. We modified the text to read:

“In other words, REM_P can be shortened but typically not extended beyond a natural limit by light stimuli, whereas SW can be both shortened and lengthened.”

There may be other stimuli that could lengthen REM episodes if more were known about the CPG process. Similar statements that need this qualification are:

Last sentence on pg 5: clarify that light is the external stimulus.

Thank you. We have now clarified (see modified text above).

Pg 10, last sentence of top paragraph: clarify that light could not increase REM duration suggesting that processes governing the switch to NREM are not affected by light.

We are not entirely confident that such a statement can be made. In conditions of sustained light pulses for example (e.g., 30 and 45s traces, Fig 2c), the termination of light is associated with a switch to SW. Similarly, the duration of SW in a long-light-pulse reset condition aligns with light-pulse termination. These results suggest that light does have a partial “inhibitory” effect on the transition to SW.

Minor point:

Fig 3b: Is the phase on the x-axis defined relative to the onset of low or high beta in the spontaneous NREM-REM pattern? Namely, how is phase = 0 defined? More information about the definition of the phase should be added to the figure caption.

We describe how phase is calculated in detail in the Methods section “Phase analyses and phase response curves”, which we have now updated with more detail as per the requests of Reviewers #1 and 2 (reviewer 2’s point 12).

In short, phase $\varphi = 0$ corresponds to the onset of low beta, that is, the transition from REM to SW. In Figure 3, in which entrainment pulses modify the period, we calculate phases in the same way.

Pg 6, bottom paragraph: “...with a widening of the phase distribution away from the optimum” This effect is not strongly visible except for the IPIs when no entrainment occurred. Maybe be more clear about what the reader should be looking for in the figure to support this statement.

Thank you for the suggestion. We updated this sentence to read:

“For IPIs causing reliable entrainment (120-200s), the distribution of pulse phases was unimodal and narrow (0.95 at 160s, corresponding to the REM_P-SW transition, Fig. 3b), indicating phase-locking of stimulus and cycle (160s, Fig. 3c). For IPIs with poor entrainment, the distribution displayed high variance (240s, Fig 3b) with pulses producing a mixture of phase advances and delays (240s, Fig 3c).”

Overall, the paper is well-written and clear. The validity of approach and quality of data are excellent. Data analyses and statistical analyses are appropriate and well done.

Many thanks. We are grateful for all these helpful comments and suggestions from the referees.

Reviewer Reports on the First Revision:

Referees' comments:

Referee #1 (Remarks to the Author):

In the revised version of the manuscript, the authors have addressed some of the comments suggested by the reviewers. However, several things remain to be addressed:

1) The impact of different sensory in the susceptibility of sleep and its different phases should be added to the discussion. Light per se is a direct modulator of circadian, therefore making this sensory modality (perhaps) different than others. The authors seem to think their results are general to all sensory stimuli so they should discuss (i.e., add to their discussion) why and how.

2) The argument about “information about population firing rates is thus implicitly contained in our metrics” is a misleading statement. The question was about whether or not single units change their firing rates during the strong manipulation (eye closure) used to test their hypothesis. In mammals, eye closing (Maffei et al, 2006, 2008; Hengen et al,, 2013...) causes changes in overall firing rates. A similar effect in *Pogona* may or may not affect their results. On the other hand, many single neurons have distinct firing rate dynamics during sleep phases independently of the overall field potential signature recorded locally. I think adding some information related to single units is necessary for making this research the most impactful in the context of existing literature.

Overall, the authors are doing a good job addressing some of the questions from the reviewers, methods and statistics are clear and easy to follow, and adding some final details will help the community understand the very important question of how sleep, and associated neural signatures, evolved.

Referee #2 (Remarks to the Author):

As described in my previous review, this manuscript explores the hypothesis that *Pogona*'s NREM-REM ultradian rhythm is produced by a central pattern generator (CPG). The authors have satisfactorily addressed my previously detailed questions and concerns. I have no further concerns.

Referee #3 (Remarks to the Author):

The revisions have addressed all my concerns

Author Rebuttals to First Revision:

Responses (II) to Referees' comments:

Referee #1 (Remarks to the Author):

In the revised version of the manuscript, the authors have addressed some of the comments suggested by the reviewers. However, several things remain to be addressed:

1) The impact of different sensory in the susceptibility of sleep and its different phases should be added to the discussion. Light per se is a direct modulator of circadian, therefore making this sensory modality (perhaps) different than others. The authors seem to think their results are general to all sensory stimuli so they should discuss (i.e., add to their discussion) why and how.

We thank the reviewer for their comments. As discussed in more detail in our previous response to rev#3, it is important to distinguish the circadian- and ultradian- rhythm mechanisms, which we do not believe to be the same.

While light clearly acts upon the ultradian rhythm generator, the precise circuits relaying light signals to the CPGs are unknown, although, as we indicated in an earlier response, we think it is unlikely that they necessitate the SCN, mainly because of the speed of their action on the CPG. We are not in a position yet to speculate about possible effects of other sensory modalities because we have not explored any. We indicate this now in our discussion with the new text:

“Whether reset and entrainment of the sleep rhythm can be obtained with other sensory modalities is thus far unknown”

2) The argument about “information about population firing rates is thus implicitly contained in our metrics” is a misleading statement.

We have previously shown that REM activity in the claustrum is characterized by the production of sharp negative field potentials (SNs) whose interval statistics match the REM-associated increase in LFP beta power (Fenk et al., 2023). SNs likely reflect net depolarizing currents in claustrum neurons, accounting for the observation that single units tend to fire action potentials aligned to the descending phase of these waveforms. We insert below Fig. 1e of our previous paper to highlight this tight relationship, and refer the reviewer to the full Fig. 1, Extended

Data Fig. 1e, and the published response to the reviewers of that paper for more detail and quantifications (Fenk et al., 2023).

Our statement was hopefully not misleading. Indeed, our previous studies (Norimoto, Fenk et al., 2020; Fenk et al., 2023) allow one to draw inferences of how the spiking probability of individual claustral neurons — and by extension population firing rates — change with the occurrence of highly stereotyped LFP features in *Pogona* (SWRs during SW, SNs during REM sleep). An increase in the rate of claustral SNs is concomitant with a rise in beta power (the primary, but not sole metric used in this paper), whether induced by light (Fig. 2a) or not. Because of the SN/spike relationship described above, information about population firing rates is implicitly contained in our metrics.

This is not to say that we have a full picture of potentially interesting spiking patterns, or of differences that may exist between individual cell-types in the claustrum (on the contrary). But also note that while the claustrum — by virtue of its convenient position in the forebrain, our detailed knowledge of its anatomy, connectivity, molecular profile, and physiology (Norimoto, Fenk et al., 2020; Fenk et al., 2023) — is an ideal recording site to identify REM/SW states and their rhythmic alternation in *Pogona*, the generation of this ultradian sleep rhythm is independent of this area (see lesion results in Fig. 4a of Norimoto, Fenk et al., 2020); it likely depends on CPG circuits located in the brainstem, upstream of the claustrum (this paper).

The question was about whether or not single units change their firing rates during the strong manipulation (eye closure) used to test their hypothesis. In mammals, eye closing (Maffei et al, 2006, 2008; Hengen et al., 2013...) causes changes in overall firing rates. A similar effect in *Pogona* may or may not affect their results. On the other hand, many single neurons have distinct firing rate dynamics during sleep phases independently of the overall field potential signature recorded locally. I think adding some information related to single units is necessary for making this research the most impactful in the context of existing literature.

We thank the reviewer for raising these points, that we hope to clarify in the following.

Our experiments, with the exception of those shown in Fig. 4, have been conducted in adult, sleeping animals, when the eyes are naturally closed and REM and SW alternate regularly (Extended Data Fig. 3b, Supplementary Video 1). In cases when we temporarily occluded one or both eyes (Fig. 5 and Extended Data Fig. 6), we attached

removable cups in the evening, around 30 min before starting an overnight recording, and at the time when the animals would usually start to slowly close their eyes and doze off. During the night, the animals were sleeping in constant darkness, interrupted only by a few brief (1s) light pulses, to test the effect of monocular stimulation (Fig. 5). The cups were removed early the next morning, suggesting minimal visual deprivation. This is very different from the manipulations used in the studies the reviewer mentions, where extended visual deprivation through monocular eyelid suture produces strong firing rate changes in the rat cortex, when long enough: for one and a half days during the critical period of developing rats (Maffei et al, 2006) and for several days in adults (Hengen et al, 2013). Moreover, and as suggested by the reviewer, we already quantified whether this manipulation resulted in a change in the statistics of REM (new Extended Data Fig. 2). From all our metrics, we only observed a very small increase in the duration of REM, within the range of no-cup, non-stimulated sleep. This increase, however, was similar to the increase observed for no-cup stimulated sleep.

For our recordings in awake animals, for which we have indeed spike-sorted a subset of the datasets, single unit activity and population averages are shown, in Fig. 4d. Periods of (“voluntary”) eye closing in awake animals, induced by turning off the lights, are accompanied by the transition into a SW-like state and the regular production of SWRs in the *Pogona* claustrum (Fig. 4b, left). Spike rate increases transiently during SWRs (Shein-Idelson et al., 2016; Norimoto, Fenk et al., 2020). However, these events occur on a background of very low activity, and at a rate that is much lower than that of claustral SNs characterizing REM (Fenk et al., 2023), consistent with the overall higher spiking rates during REM-like states (lights on) (Fig. 4d). No eye cups were used in our awake experiments.

Regarding firing rate dynamics of individual neurons, for different sleep phases, and their possibly differential association with the simultaneously recorded LFP: for the areas we have been recording from, such as the DVR (Shein-Idelson et al, 2016), the claustrum (Norimoto, Fenk et al., 2020; Fenk et al., 2023), or the isthmic nucleus lmc (Fenk et al., 2023), firing rates are typically well-predicted by the field potential recorded locally (and *vice versa*). That is, claustral spiking activity increases transiently around the occurrence of SWRs during SW, and around the time of SNs during REM sleep. lmc activity is very low during SW sleep (also characterized by the absence of SWRs in this region), and very high during REM, when spiking is again tightly locked to the occurrence of sharp negative waveforms in the LFP (Fig. 4h and Extended Data Fig. 7g in Fenk et al., 2023). These observations do not rule out the existence of cell types that deviate from this general pattern, and we do in fact have some preliminary evidence that such cells exist. However, neither do we currently have sufficient data to further characterize the properties of these rare or undersampled neurons, nor would it change any of the main conclusions of this paper.

In summary, while there clearly remains a lot to explore regarding spiking activity in the claustrum, such as the electrophysiological properties and behaviour of molecularly defined cell-types (Norimoto, Fenk et al., 2020), answering these questions would require targeted and entirely new experiments that are beyond scope of this paper.

Overall, the authors are doing a good job addressing some of the questions from the reviewers, methods and statistics are clear and easy to follow, and adding some final details will help the community understand the very important question of how sleep, and associated neural signatures, evolved.

We thank the reviewer for their constructive comments, and hope to have addressed their remaining questions.